# Long-term series and trends in surface solar radiation at Athens,
# Greece
Stelios Kazadzis[1,2], Dimitra Founda[2], Basil E. Psiloglou[2], Harry Kambezidis[2], Nickolaos
Mihalopoulos[2,3], Arturo Sanchez-Lorenzo[4,5], Charikleia Meleti[6], Panagiotis I. Raptis[1,2], Fragiskos
Pierros[2], Pierre Nabat[7]
[1] {Physikalisch-Meteorologisches Observatorium Davos, World Radiation Center (PMOD/WRC)
Dorfstrasse 33, CH-7260 Davos Dorf, Switzerland}
[2] {Institute of Environmental Research and Sustainable Development, National Observatory of Athens,
Greece}
[3] {Department of Chemistry, Univ. of Crete, Heraklion, Crete}
[4] {Instituto Pirenaico de Ecología, Consejo Superior de Investigaciones Científicas (IPE-CSIC), Zaragoza,
Spain}
[5] {Department of Physics, University of Extremadura, Badajoz, Spain}
[6] {Physics Department, Aristotle University of Thessaloniki, Greece}
[7] {CNRM UMR 3589, Météo-France/CNRS, Toulouse, France}
Corresponding author: S. Kazadzis, kazadzis@noa.gr

# 1  **Abstract**

We present a long-term series of solar surface radiation (SSR) from the city of Athens, Greece. SSR
measurements were performed from 1954 to 2012, and before that (1900-1953) sunshine duration
(SD) records have been used in order to reconstruct monthly SSR. Analysis of the whole dataset
(1900-2012) mainly showed very small (0.02%) changes in SSR from 1900 to 1953, including a
maximum decrease of 2.9% per decade in SSR during the 1910 to 1940 period, assuming a linear
change. For the dimming period 1955-1980, a -2% change per decade has been observed that
matches various European long-term SSR measurement related studies. This percentage at Athens
is in the lower limit, compared to other studies in the Mediterranean area. For the brightening period
1980-2012 we have calculated a +1.5% change per decade, which is also in the lower limit of the
reported positive changes in SSR around Europe. Comparing the 30-year periods 1954-1983 and
1983-2012, we have found a difference of 4.5%. However, measurements of the first 30 year period
are associated with higher uncertainties than those of the second period, especially when looking at
year to year changes. The difference between the two periods was observed for all seasons except
winter. Analyzing SSR calculations of all sky and clear sky (cloudless) conditions/days, we report
that most of the observed changes in SSR after 1954 can be attributed partly to cloudiness and
mostly to aerosol load changes.

# 1 Introduction

In the past decades surface solar radiation (SSR) and the transmission of the atmosphere have been of increasing interest because of the related impacts on climate. Most of the energy in the Earth-atmosphere system is introduced by solar radiation. It provides heating, which creates pressure gradients and ultimately wind and triggers water, carbon and oxygen cycles through evaporation and photosynthesis. These processes define the climatological conditions, and changes of incoming solar radiation rapidly affect the energy balance (Wild et al., 2015). Interest in solar radiation changes has also been raised after the development of solar energy applications, which are continuously growing in number over the recent years. Changes in SSR have been recorded over the last century and can be caused either by natural events such as volcanic eruptions or human-related activities, mainly in polluted regions (Wild, 2016). At longer scales (thousands of years) changes in SSR might have been caused by changes in the Earth's orbit and Sun solar output (Lean, 1997; Ohmura, 2006).

Systematic continuous measurements of SSR were established in the middle of the 20th century at selected meteorological observatories. Solar variations have been investigated in several studies using ground based SSR measurements from various monitoring networks worldwide (e.g., Ohmura, 2009) and also by satellite-derived estimations (e.g. Kambezidis et al., 2010). Overall, most of these studies (Gilgen et al., 1998; Noris and Wild, 2009; Wild, 2009 and 2016 and references therein) have reported a worldwide decrease of solar incoming radiation in the period 1960-1985 (known as dimming period), followed by an increase (brightening period) thereafter. These changes were reported to be higher in more polluted and urban areas but have also been recorded in isolated regions such as the Arctic (Stanhill, 1995) and Antarctica (Stanhill and Cohen 1997). Other recent studies have investigated the effect of urbanization on global brightening and dimming, and found no marked differences among urban and rural SSR time series (Tanaka et al., 2016) and Imamovic et al. (2016). Changes in atmospheric transmission due to variations in cloudiness and aerosol concentration are the main factors to be investigated in order to determine the possible causes of such trends in SSR (Wild, 2009). However, due to the aerosol-cloud interactions and the aerosol indirect effect on SSR, the two factors (clouds and aerosols) are not completely mutually exclusive in explaining SSR changes.

The cloud and aerosol radiative effects on solar radiation variations over the past decades have been investigated by numerous studies during the last years. The inter-annual variations in cloudiness is crucial for studying SSR time series, but its decadal variability is not always connected with the widespread dimming and brightening effects (Wang et al., 2012; Wild, 2016). Aerosols play a

significant role in incoming radiation, by scattering and absorbing light and by acting as cloud-condensation nuclei. Over the 20-year dimming phase (from 1960 to 1980) and the 15-year brightening phase (from 1990 to 2005), it was found that the aerosol effects (direct and indirect) played the most important role in SSR variation (Dudok de Wit et al., 2015). Concerning Central Europe, Ruckstuhl et al. (2008) suggested that the brightening phase under cloud-free conditions is in line with decreasing anthropogenic aerosol emissions (Streets et al., 2006). Nabat et al. (2013) using a blending of remote sensing and model products showed that a decreasing Aerosol Optical Depth (AOD) trend of 0.05 per decade in Europe for the period of their study (1979-2009). In addition, Nabat et al. (2014) reported that anthropogenic aerosol decline in Europe from 1980 to 2012 statistically explains $81 \pm 16\%$ of the observed brightening. Overall, changes in anthropogenic aerosol emissions are now considered as the major cause of brightening and dimming effects (Wild, 2016). The gaseous and particulate air pollutants may reduce solar radiation by up to 40% during air pollution episodes (Jauregui and Luyando, 1999). Aerosol related attenuation is much larger during forest fires, dust events and volcanic eruptions. Vautard et al. (2009) have also reported a decline of the frequency of low-visibility conditions such as fog, mist and haze in Europe over the past 30 years, suggesting a significant contribution of air-quality improvements

Long-term series of SSR measurements are essential for such studies. One of the main constraints in studying SSR temporal changes is the small number of sites with reliable long-term records, even over areas with a relatively high density of stations such as Europe, Japan or the USA. In Europe for example, there are currently less than 80 stations with more than 40-years homogeneous data (Sanchez-Lorenzo et al., 2015), with very few of them operating over Southern Europe. Recently, a high-quality dataset of SSR has been set up over Italy (Manara et al., 2016), but there is still lack of high quality long-term trends in other countries around the Mediterranean Basin.

In addition, even more sporadic measurements are available before the 1950s (Stanhill and Ahiman, 2016); the few studies of them have pointed out an SSR increase in the first decades of the 20[th] century and a maximum around 1950 (Ohmura, 2006 and 2009). This topic is still controversial due to the few long-term series available (Antón et al., 2014). Recently, there have been efforts to reconstruct SSR series in periods with no direct measurements available, using other variables such as sunshine duration (SD), which is available in a large number of sites since the late 19[th] century (e.g., Stanhill and Cohen, 2005, for USA; Sanchez-Lorenzo and Wild 2012, for Switzerland; Matuszko 2014, for Poland). For example, Sanchez-Lorenzo and Wild (2012) used data from 17 stations in Switzerland, considered SD as a proxy and successfully reconstructed SSR time series since the late 19[th] century. They calculated that the variability in SSR monthly anomalies can be

explained by SD anomalies in a range of 76%-96%, and a monthly root mean squared error of 4.2 W m$^{-2}$ between recorded and estimated SSR for all-sky conditions and of 5.5 W m$^{-2}$ for clear-sky conditions. Other studies have tried to use pan evaporation as a proxy of SSR, for the first half of the 20$^{th}$ century (Stanhill and Möller, 2008). Kambezidis et al. (2016) used monthly re-analysis datasets from the Modern Era Retrospective-Analysis for Research and Applications (MERRA) and calculated shortwave radiation trends over the period 1979-2012 for the Mediterranean basin. They reported an increase in MERRA of +0.36 W m$^{-2}$ per decade, with higher rates over the western Mediterranean (+0.82 W m$^{-2}$ per decade).

A few studies discuss the brightening/dimming effect in the southeastern Mediterranean. Zerefos et al. (2009) have studied the Ultraviolet A (UVA) changes for the area of Thessaloniki (Greece) from 1984 to 2008. They calculated a 5% positive trend per decade linked to a negative trend in aerosol optical depth (AOD) for the area due to air pollution abatement strategies. The variability in shortwave downward solar irradiance received at Earth's surface over Thessaloniki, Greece, for the period 1993-2011 (Bais et al., 2013), showed an upward trend in SSR after 1990 (+0.33%  per year). They also reported signs of a slowdown in the upward trend in SSR during the beginning of the 2000s. Founda et al., (2014) have studied the SD long-term variability over Athens area. They reported a 7% decline in the annual SD from 1951-1982 and a 3% increase from 1983-2011 under all sky conditions. Under near clear sky conditions, these percentages are -7% and + 9% for the dimming and brightening periods respectively. Similarly, Founda et al. (2016a) analyzed long-term SD and total cloud cover time series over 15 sites in Greece (the oldest one beginning in 1897). They have shown an increase in SD almost at all stations since the mid-1980s, which in certain areas of Southeastern Greece amounts to an increase of 20 h per year. This increase is not accompanied with synchronous decrease in total cloud cover, possibly evidencing to decreasing aerosol loads, despite the fact that their impact on SD should be lower than on SSR (Sanchez-Romero et al., 2014). Yildirim et al. (2014) have analyzed 41 years of SD measurements in 36 stations in Turkey. They reported a decreasing trend (between 1970 to about 1990) at most of the stations. After 1990, they observed either zero trend variation or a reduction in the decreasing rate of SD at most of the locations. They concluded that the decreasing period might be attributed to human-induced air pollution. Founda et al. (2016b) have investigated the visibility trends over Athens area from 1931 to 2013. They reported a deterioration in the visibility up to 2004 and a slight recovery afterwards, negatively/positively correlated with relative humidity/wind speed and positively correlated with AOD from 2000 to 2013. Finally, Alexandri et al., 2017 studied the spatio-temporal variability in SSR over the eastern Mediterranean for the 1983–2013 period, using the Satellite Application Facility on Climate Monitoring Solar surfAce RAdiation Heliosat satellite-

based product (SAF). They reported a positive (brightening) and statistically significant SSR trend at the 95% confidence level ($0.2 \pm 0.05$ W m$^{-2}$ year$^{-1}$ or ($0.1 \pm 0.02\%$ year$^{-1}$) being almost the same over land and sea.

In this work, measurements of SSR, recorded over 60 years at the center of Athens, are presented. In addition, with the use of the SD measurements that are conducted in Athens since 1900, we could reconstruct the time series of SSR during the first half of the 20th century. These time series (1900-2012) are the oldest, uninterrupted and high quality SSR time series in the SE Mediterranean and one of the oldest in Europe, providing unique information about the variations and trends in the area for the past decades. Time-series of SSR over Athens are presented to try answering questions such as:

Are the dimming–brightening patterns observed in Europe over the past century also observed, at the same extent, over the eastern Mediterranean?

Is SSR variability during the first decades of the 20th century in Athens in line with the trends reported at other locations over this period?

Can we verify that anthropogenic aerosols play the most important role on the brightening/dimming observed SSR after 1950, in agreement with results from other European regions?

## 2 Data and Methodology

### 2.1 DDR data collection and analysis

The SSR data used in this study cover the period from December 1953 to December 2012 and were measured by a series of pyranometers that are mentioned in Table 1. These instruments have been operating continuously at the Actinometric Station of the National Observatory of Athens (ASNOA) (Hill of Pnyx, Thissio), that is located near the center of Athens, Greece (38.00$^{\circ}$ N, 23.73$^{\circ}$ E, 110 m above mean sea level). Table 1 presents the instruments and the period of operation, as well as the maximum error on the integrated daily values. References mentioned in Table 1 describe the exact type of errors and uncertainties related to the sensors. In the period 1953-1986, the maximum daily error was about 5%, and 2% afterwards. The spectral response of the sensors is in the range of 285-2800 nm; since 1986 a first-class Eppley PSP pyranometer (WMO, 1983) is operating at ASNOA. Since 1992, frequent calibrations (every two years) have been performed by the NOA's Laboratory of Meteorological Device Calibration (LMDC, 2016) in order to ensure the high quality of measurements. LMDC follows the standard calibration procedure for thermopile pyranometers (ISO 9847, 1992), with exposure to real sunlight conditions and

comparison with a standard thermopile pyranometer (Secondary Standard). LMDC's reference
pyranometer, Kipp & Zonen CM21, is regularly calibrated in PMOD/WRC, Davos, Switzerland.
Table 1: History of SSR instruments used at ASNOA. SSR measurements refer to the total solar
radiation on a horizontal surface.

| | Instrument | Period | Class | Maximum error (daily integral) | Reference | Class | Comments | Resolution |
|---|---|---|---|---|---|---|---|---|
| 1 | Solarigraph GOREZYNSKI | 1953-1959 | 2nd | 5% | Coulson (1975) | B | One instrument being used | 1 hour |
| 2 | Eppley 180º pyranometer (No. 3604) | 1960-1966 | 2nd | 5% | Coulson (1975), Drummond (1965) | B | Manual measurements archiving with mvoltometer | 1 hour |
| 3 | Eppley 180º pyranometer (No. 3604) coupled with a Leeds-Northup recorder, Speedomax, type G | 1966-1968 | 2nd | 5% | Coulson (1975), Drummond (1965) | B | Same instrument as #2 with Speedomax recorder | 1 hour |
| 4 | Eppley 180º pyranometer (No. 3034) coupled with a Leeds-Northup recorder, Speedomax, type G | 1968-1973 | 2nd | 5% | Coulson (1975), Drummond (1965) | B | New instrument, same recorder | 1 hour |
| 5 | Eppley pyranometer, type 8-48 and type 8-48A coupled with a Leeds-Northup recorder, Speedomax, type G | 1974-1986 | 2nd | 3-5% | Hulstrom (1989) | B | Type 8-48 and type 8-48A instruments were measuring alternatively for three years each | 1 h |
| 6 | Eppley Precision Spectral Pyranometer (PSP) | 1986-now | 1st | 1-2% | Hulstrom (1989) | A | Regular recalibrations. Coupled with a, A/D recorder (Campbell Scientific Ltd.) Datalogger, type CR-21X at the beginning until 2003, a CR10X until 2012 | 1 min |

SSR data are processed using a set of quality-control (QC) tests in order to ensure the quality of the
data set. The QC procedures include rejection of:
• Measurements  for solar elevation angles less than 5 degrees;

- SSR values equal to or less than 5 W m$^{-2}$, during sunrise and sunset, due to the pyranometers' sensitivity;

- SSR values greater than 120% of the seasonally corrected solar constant.

After the initiation of diffuse horizontal radiation measurements at ASNOA in 1991, the following quality criteria were added for rejection:

- diffuse horizontal values greater than the corresponding SSR ones;

- diffuse horizontal values greater than 80% of the seasonally correct solar constant;

- direct-beam solar component exceeding the extraterrestrial solar irradiance.

Also, both total and diffuse horizontal measurements are corrected for the night-time dark-signal offset of the pyranometers.

Mean daily SSR values were calculated from the data set of this study (December 1953 – December 2012); only months with more than 20 days of measurements were considered in the analysis. Over the 60 years of measurements, only three months (January and February of 1998 and March of 2012) did not fulfill this criterion.

When trying to use such long term series, it is evident that the data quality differs as instrument performance have been improved, quality assurance and quality control procedures have been standardized, and finally the information flow on the day to day instrument performance issues is much more complete in the recent time. At ASNOA, (after 1986 only) the instruments were calibrated or checked with a reference instrument on a yearly basis to identify changes in the calibration and drifts. As reported, the addition of diffuse irradiance measuring instruments provided the opportunity to improve also minute based measurement quality. Before 1986, the instruments reported in table 1 have been used. According to the log books there has always been a certain overlap when changing from one instrument to another. Reports mention that there were instrument drifts that have been corrected with no further information from 1953 to 1970. Instrument overlaps after 1986 were used to eliminate possible instrument related offsets. However, instrument differences (e.g. thermal offset of PSP instrument compared with 8-48 pyranometer, Vignola et al., 2016) could theoretically have an effect in the order of 1-2 W m$^{-2}$ on the series continuation. In this case the subtraction of the night time dark signal (more specific the mean of the previous and next night signal was subtracted for a specific day) reduces at least in half the problem. However, in order to answer to the reviewer question, the remaining offset was not considered in our analysis, as partly it have been tackled through the overlapping measurements/homogenization procedures. In addition, the inclusion of diffuse radiation in the

quality assurance tests after 1991 could cause a major improvement on the newest data compared
with the old ones. However, this recent improvement in quality control cannot be linked with a
systematic impact on SSR measurements and changes compared to the past, other than higher
uncertainty on the integrated (monthly, yearly) SSR values. For the 1953-1986 time series there is a
number of publications that have been using the SSR-NOA time series. More specific: Macris,
(1959), has used the 1954-1956 SSR measurements to identify the relationship between SSR and
sunshine duration. Katsoulis and Papachristopoulos, (1978), have used the SSR data from 1960 to
1976 in order to calculate SSR statistics for daily, seasonal and yearly solar radiation levels at
Athens, Greece. Notaridou and Lalas, (1979), have used the 1954-1976 SSR data in order to verify
an empirical formula on global net radiation over Greece. Flocas, (1980) has used the 1961-1975
SSR time series to compare them with sunshine duration data for the same period. Kouremenos et
al., (1985) have used the SSR data from 1955-1980 in order to correlate their changes with various
atmospheric parameters. Zabara, (1986) has used the 1965-1980 time series to verify a monthly
solar radiation calculation method. Katsoulis and Leontaris, (1981), have used the 1960-1977 data
to verify tools describing the solar radiation distribution over Greece. Finally, the percentages of
errors reported in table 1 are not directly linked with possible instrument drifts that can impact the
SSR time series analysis. So results of measurements before 1986 have to be used with caution and
accompanied by a report on the different level of uncertainties of the past and recent data.
Figure 1 shows the intra-annual variability of SSR at ASNOA based on the measurements from all
instruments during the period 1953-2012. Daily SSR at Athens ranges between approximately 6 to
27 MJ m$^{-2}$ during the year. Mean and standard deviations were calculated using the 60 year record
for each day.

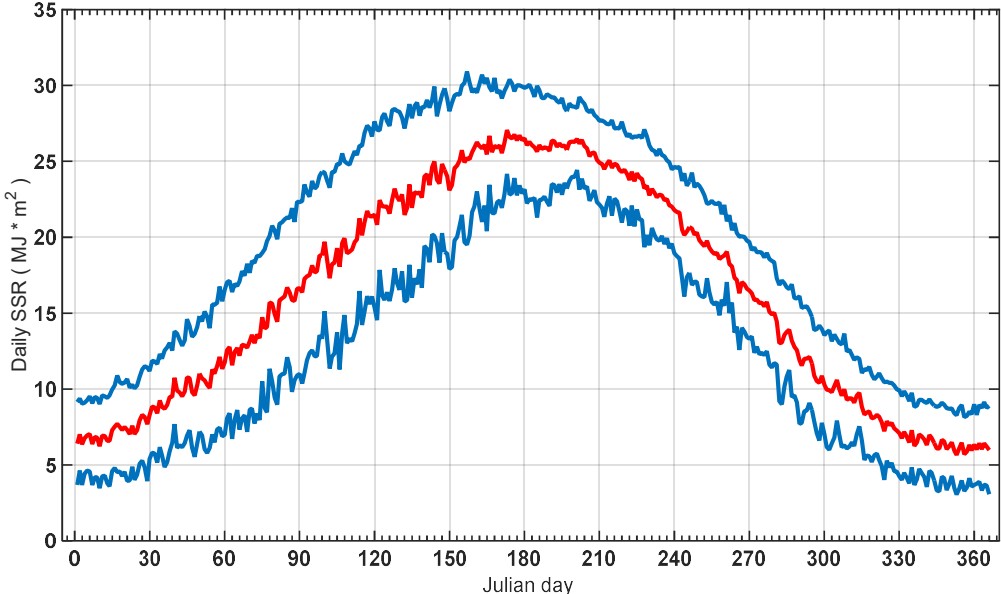

**Figure 1.** Average intra-annual variability of Surface Solar Radiation (SSR) at Actinometric Station of the
National Observatory of Athens (red), along with the inter-annual variability for a given day (±1 standard
deviation, blue), calculated over the period 1953-2012.
The results of figure 1 show the average yearly pattern of SSR at ASNOA. The day to day
variability that is shown as "noise" in the plotted blue line comes from the 60 year averaging of
each day and is mostly related with the amount of cloudiness for each of the averaged days.
Minimum and maximum SSRs at solstices, compared to a cloudless sky aerosol free model, are also
related with the highest probability of the presence of clouds during winter months. For the
calculation of each of the daily averages the available data points vary from 55 to the possible
maximum of 60.
Concerning the data availability for SSR and SD data, SSR monthly means calculated here have
been retrieved from daily calculated SSRs. Over the 59 years (708 months) of SSR data, 98% of the
months had none or one day missing, 3 months had from 10-20 missing days and 2 months from
20-30 missing days.  For SD, 1931-1940 monthly data used have been taken from the NOA
measurement annals. From 1940 on, hourly measurements have been used in order to derive daily
and monthly measurements. The SD time series have no gaps with only six missing days during
December 1944 (Founda et al., 2014).
**2.2 Sunshine duration data**
In addition, collocated measurements of SD recorded at ASNOA have been used. According to
WMO (2010), the SD during a given period is defined as the sum of the sub-periods for which the
direct solar irradiance exceeds 120 W m$^{-2}$. In Athens, SD has been recorded using classical
Campbell-Stokes heliographs (since 1894) and been replaced by electronic instrumentation in 1998
(EKO, MS-091 analog SD sensor). Monthly SD values since January 1900 have been used in this
study. A more analytical study of these time series can be found in Founda et al. (2014).
Complementary to this study, cloud-cover observations from the Hellenic National Meteorological
Service (HNMS) from 1954 have also been used. These observations are recorded at a site 7 km
away from ASNOA. All cloud observations at HNMS are conducted every 3 hours and are
expressed in octas.
**2.3 Aerosol Optical Depth (AOD)**
In order to examine the AOD impact on SSR, we have used the longest satellite based AOD series
available for the area. This is the AOD time series from Advanced Very High Resolution
Radiometer (AVHHR). AOD retrievals at 630 nm over global oceans at spatial resolution of 0.1$^{o}$ x
0.1$^{o}$ and one overpass per day have been used. Data used were downloaded from NOAA Climate
Data Record (CDR) version 2 of aerosol optical thickness (Zho and Chan, 2014), and cover the
period from August 1981 to December 2009. AVHHR AOD embodies a large variety of
uncertainties, including radiance calibration, systematic changes in single scattering albedo and
ocean reflectance (Mishchenko et al, 2007). Current dataset radiances have been recalibrated using
more accurate MODIS data (Chan et al, 2013). We used daily data at the region around Athens
(longitude: 37.5$^{o}$-38.2$^{o}$N, latitude: 23.2$^{o}$-24.4$^{o}$E) which includes 50 active available (ocean) grid-
points. The above region was selected based on data availability on each grid, within 50 km from
ASNOA.
To complement the analysis on the evolution of aerosols, the recent climatology developed by
Nabat et al., (2013) has been considered over the period 1979-2012. This product provides monthly
averages of AOD at 550 nm over the Mediterranean region at 50 km resolution. It is based on a
combination of satellite-derived (MODIS instrument) and model-simulated products (MACC
reanalysis and RegCM-4 simulations), which have been selected among many available datasets,
from an evaluation against ground-based measurements of the AERONET network. Thus this
climatology is able to give the best possible atmospheric aerosol content over the period 1979-2012.
For the present work, the AOD time series over the grid cell of the ASNOA (38.00$^{o}$ N, 23.73$^{o}$ E)
has been extracted and is referred to as the ChArMEx data thereafter.
**2.4 Clear-sky SSR**
For the determination of the clear sky (defined here as the cloudless) days, we have used both the
cloud octas and SD data. Daily observations have been used for this analysis. We have defined as a
clear sky day each day that fulfills the following criteria:
-    the mean daily cloudiness (in octas) should be less than 1.5, and
-    the total daily SD should be higher than 90% of its theoretical (astronomical) value.
The procedure for calculating a single mean cloud octa value for each day was the following:
We have first excluded night-time cloud observations; then, we have weighted each observation
based on the hour of the observation. Weights have been calculated based on the solar radiation
contribution of the specific time slot and day of the month, compared with the daily clear sky SSR
integral, of the particular day and month.
**2.5 Reconstruction of SSR from SD**
We have used the 1900-2012 SD time series in order to extend our SSR time series back to 1900.
There are different methods that are used in order to estimate SSR values from SD. In this work we
have tried two methods based on the linear regression between SSR and SD (Sanchez-Lorenzo and
Wild, 2012). For all-sky conditions the monthly anomalies, obtained as differences from the 1983-
2012 mean, of SSR and SD have been calculated. Then for each month a linear regression has been
used to estimate the relationship between the SSR and the SD:
**SSR = $a$ * SD + $b$**                          (1)
Figure 2 shows four out of the total twelve regressions together with its' statistics.

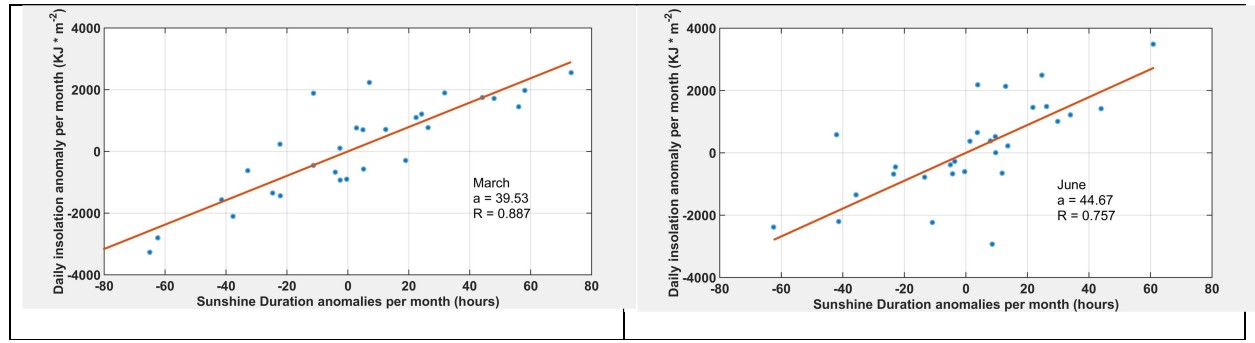

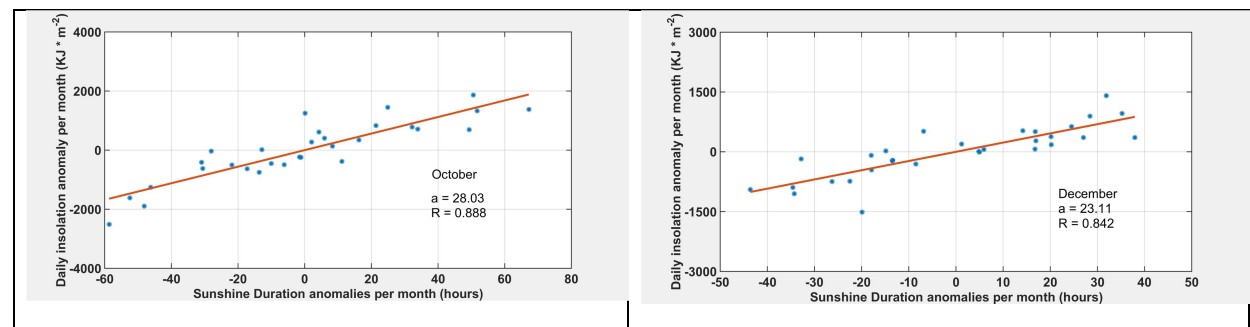

Figure 2. Linear regression of SSR and SD anomalies for March, June, October and December,
2    1983-2012.

Table 2. Monthly regression statistics of SSR vs SD anomalies (see eq. 1).

| month | Jan | Feb | Mar | Apr | May | Jun | Jul | Aug | Sep | Oct | Nov | Dec |
|-------|-----|-----|-----|-----|-----|-----|-----|-----|-----|-----|-----|-----|
| $a$ | 22.47 | 34.99 | 39.53 | 46.65 | 57.88 | 44.67 | 51.87 | 46.23 | 34.28 | 28.02 | 27.32 | 23.10 |
| $R$ | 0.842 | 0.895 | 0.887 | 0.840 | 0.799 | 0.757 | 0.773 | 0.772 | 0.812 | 0.888 | 0.916 | 0.842 |

The correlation coefficients ($R$) they vary from 0.75 to 0.91. This implies that the SD monthly
anomalies explain between 56% and 83% of the variability of the  SSR  monthly  anomalies. It has
to be noted that coefficient $b$ is less than $10^{-3}$ for all months.
The second method to derive a relationship between SD and SSR, was based on the broadly
accepted formula of Ångström (1924):
**SSR/SSR$_{\mathbf{max}}$=c + d * (SD/SD$_{\mathbf{max}}$)**                              **(2)**
where SSR$_{max}$ and SD$_{max}$ refer to the theoretical extra-terrestrial value of SSR and the astronomical
value of SD, respectively, while $c$ and $d$ are constants usually defined monthly. This formula can
only be used in large data sets as a statistical approach. That is because for different cloud heights,
thicknesses and positioning, the constants can show a large variability (Angell, 1990). The 1983-
2012 period was chosen for determining the SSR vs. SD relationship as SSR measurements have
lower uncertainties compared with the 1953-2012 period. We thus calculated $c$=0.237 and $d$=0.458
and an $R^2$ equal to 0.81.
We also followed the same procedure to calculate the coefficients of the Ångström formula
separately for each month and for each season during the control period 1983-2012. For individual
months, calculated SSR/SSRmax vs SDU/SDUmax coefficients of determination ranged from 0.5
to 0.65 for winter months, 0.32 to 0.67 for spring months, 0.47 to 0.53 for autumn months and 0.1
to 0.38 for summer months. So coefficients of determination using the monthly based data were

much lower than the first reported method. The low coefficients for the summer period are related with the small range of values of SDU/SDU max and SSR/SSRmax that are related with the absence of clouds.

We have used both the monthly regression coefficients from the first method and the yearly based Ångström formulas in order to investigate the impact of the different methods on the SSR reconstruction. Results of the reconstructed SSR yearly values from 1900-1953 showed maximum differences of 1% in the calculated SSR percent anomalies, while for monthly values the higher difference was 2%. In order to avoid the use of theoretical normalization values such as SDUmax and SSRmax needed for the second method, we have reconstructed the SSR time series based on the monthly based results of the first method as proposed in Sanched-Lorenzo and Wild, 2012.

## 3 Results

### 3.1 Long-term variations and trends (1900-2012)

Based on methods described in Section 2, we have reconstructed monthly SSR from 1900 to 1953. Using the full dataset of reconstructed (1900-1953) and measured SSR (1954-2012) we have calculated the mean monthly SSR values and used them for de-seasonalising the results shown in Figure 3. The de-seasonalizing was determined by: a. calculating the average SSR (SSR$_{mi}$) for each month (i) out of the 12 months of the given year, for all 1983-2012 years, b. calculating the changes in % in SSR (SSR%(i,y)) for each month (i) of each year (y) as:

$$SSR\%(i,y) = \frac{SSRiy - SSRmi}{SSRmi} * 100 \qquad (3)$$

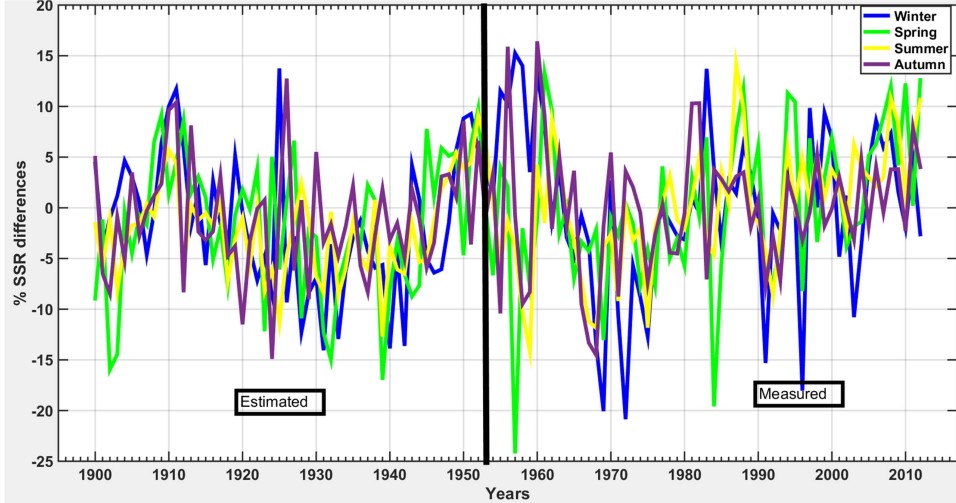

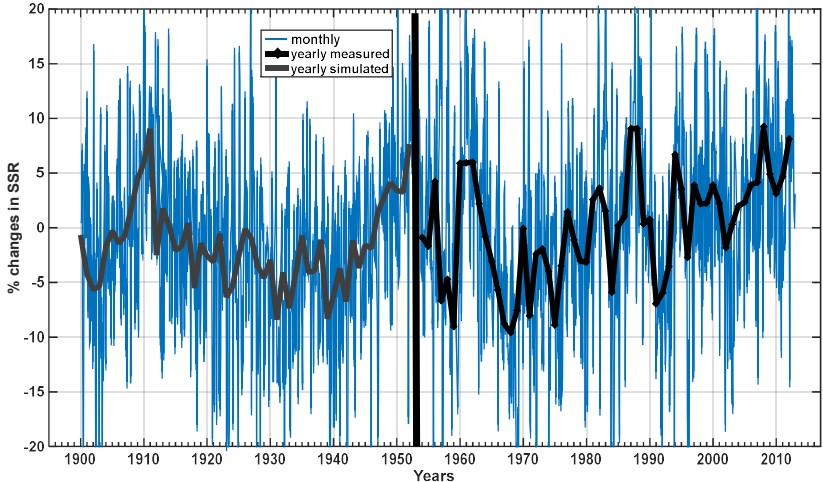

**Figure 3.** Full time series of de-seasonalised SSR percent changes (using the 1900-2012 monthly
averages). Upper panel: different colors represent seasonal analysis, lower panel: black bold line
represents the annual series, and light blue line the mean monthly values
According to Figure 3, the month-to-month variation (shown with light grey line) can reach more
than 30% in comparison with the mean monthly average of the whole data set. Annual means show
a 10%-12% (peak to peak) decrease in SSR from 1910 to late 1930's and then an increase of 12%
from 1940 to early 1950's. The simulated SSR results follow the observed decline of SD reported in
Founda et al., (2014), where a decrease from 1910 to 1940's is shown.
Subsequently, there is a decrease during the late 1950's and then a positive change of the order of
20% till today with an episode in the early 1990's that shows low SSR values. Measured SSR in
1991-1993 period differs by 5% compared with the one in 1990. The latest can be linked with the
Pinatubo volcanic eruption and its known effect in the SSR (e.g. Zerefos et al., 2012).
Analytical linear trends of each of the sub-periods and for every season are presented in Table 2. It
has to be noted that the trend determination and its statistical significance do not take into account
measurement or SSR reconstruction related uncertainties, which are different for the different
periods.
**Table 2.** Annual and seasonal SSR trends in percent per decade over the period 1900-2012 and
different sub-periods. Percentages in parenthesis show the limits of the 95% confidence bounds.

| Season | 1900-2012 | 1900-1952 | 1953-1982 | 1983-2012 |
|---|---|---|---|---|
| **Winter** | -0.11 (±0.47) | -0.90 (±1.46) | -6.43 (±3.83) | +0.52 (±3.26) |
| **Spring** | +0.54 (±0.37) | +0.38 (±1.12) | -0.60 (±3.10) | +2.77 (±3.10) |

| | | | | |
|---|---|---|---|---|
| **Summer** | +0.59 (±0.21) | +0.28 (±0.48) | -1.14 (±2.90) | +1.38 (±2.55) |
| **Autumn** | +0.21 (±0.44) | +0.11 (±0.97) | -1.28 (±3.42) | -1.50 (±1.83) |
| **Year** | **+0.39** (±0.22) | **+0.04** (±0.71) | **-2.33** (±2.28) | **+0.80** (±1.96) |

Looking at the 1900-2012 period the seasonal and annual linear trends in SSR are less than 1% per
decade. A positive change of 0.39% per decade has been calculated from annual values. For the
whole data set, all seasons show positive trends, except winter. For the periods with simulated SSR
values (1900-1952), even smaller trends have been detected for spring and summer. The measuring
period of 1954-2012 has been split into two sub-periods of 1954-1982 and 1983-2012. The first
sub-period shows a negative annual change of -2.33% per decade in SSR, which is also reflected in
all seasons with predominant changes during winter (-6.43% per decade). The second sub-period
shows a positive trend of +0.80% per decade with the highest ones in spring (+2.77% per decade)
and summer (+1.38% per decade) and negative in autumn (-1.50% per decade). Looking at the
trend significance described by the 95% confidence bounds, we can see significant positive trends
for 1900-2012 (yearly, summer and spring) and significant negative trends for yearly analysis and
winter of 1953-1982.
In order to have a better understanding of the SSR changes over the 113-year period (1900-2012),
we have calculated the decadal SSR trends for different time-windows (15 to 40 years). Figure 5
shows the results of this analysis.

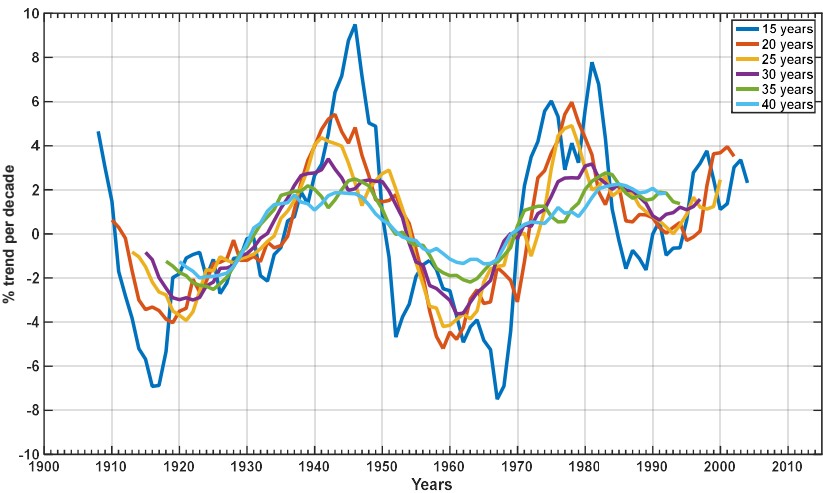

**Figure 4.** Trends in SSR (% per decade) calculated for different sliding time windows. The value of
the trend has been calculated at the central year of each time window.
For the first two decades of the 20[th] century there appears a decrease in SSR, in line with other
long-term SD series as recently shown by Stanhill and Ahiman (2016). Then, in all calculations an
increase is shown from mid-1930's to late 1940's, in line with the early brightening effect pointed
out by other authors (Ohmura, 2009; Sanchez-Lorenzo et al., 2008). It should be reminded that this
period is based on estimations of SSR from SD measurements, which thus include additional
uncertainties. Nevertheless, early dimming and brightening periods have been reported in Stanhill
and Ahiman (2016). The results can be partly supported by trends in anthropogenic black carbon
(McConnell et al., 2007; Lamarque et al., 2010) and biomass-burning (Lamarque et al., 2010)
emissions in Europe. The dimming period from 1950's to 1970's can be observed in all time
windows with a brightening effect after late 1970s.
The 40-year and 30-year time windows in the analysis presented in Figure 5 show the maximum
rate of increase in early 1940's (resulting in an increase of 2% per decade and 3% per decade,
respectively). Then a maximum rate of decrease is observed in early-mid 1960's, followed by a
positive rate of increase after 1990's. Shorter time windows (15 years) are also interesting as they
are able to capture the Pinatubo effect in early 1990's.

## 3.2 Variations and trends in SSR for the 1954-2012 measurement period

In order to further analyze the whole 59-yr SSR data set of this study, we have divided it in two 30-
yr climatological sub-periods: 1954-1983, and 1983-2012 (the common year is meant to have equal
(30 full years) duration for both periods). Investigating a possible seasonal dependence, the relative
difference in SRR for every month from its mean monthly value over the whole measurement
(1954-2012) period was calculated.
Figure 5 shows the mean daily insolation for each month for the two sub-periods and the whole 59-
year period. Examining the monthly average differences between the two periods, we observe that
for spring and summer months these are of the order of 6%. In addition, for all months SSR
differences of the 1983-2012 period compared to the 1954-1983 period are positive with an
exception of November (-1.9%) and December (-1.2%). In general, the second measurement period
shows a 3% to 8% larger monthly SSR than the first measurement period.

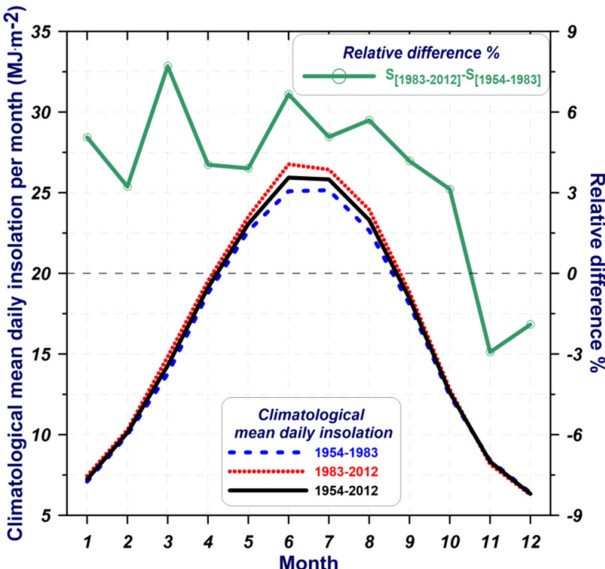

**Figure 5.** Intra-annual variability of monthly mean daily SSR over the sub-periods of 1954-1983
(blue line) and 1983-2012 (red line) and the entire period of 1954-2012 (black line). The green line
(right axis) represents the monthly relative difference between the two 30-year sub-periods (recent
minus older period).

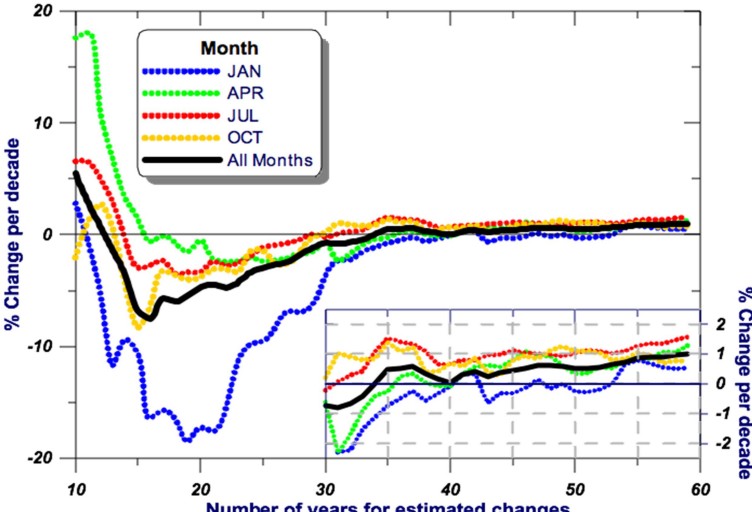

**Figure 6.** Percent change per decade for different time scales and different months using 1954 as
the starting year (the last 30 year period is magnified with changes presented as percent per
decade).
We have also calculated decadal trends in time windows of 15 to 30 years for the entire SSR
measurement period (see Figure 6), only for the 1954 to 2012 period, Figure 6 shows the SSR
change per decade for the months of January, April, July, October and yearly (all months). The
figure is showing a trend analysis for the entire data set with time windows from 10 to 59 years,
where each time window starts from 1953. For all months SSR changes become positive for time
windows of 35 years and higher (1953-1988 time window and any larger window starting from
1953). Negative trends calculated from 1954 to any given year up to 1989 are mainly due to the
large negative changes during the winter period. Especially during the 1954-1974 period, winter
SSR changes show a 18% per decade decrease. Linear trends in SSR from 1954-2012 showed a
positive trend of the order of 1% per decade, while individual months vary from 0.5% per decade to
1.5% per decade. Mostly positive trends are detected using any time window centered after 1975.
Larger trends are calculated for time windows centered at 1975 to 1980 and after 2000 (in the order
of 5% per decade using the 15-year time window). For the period 1954 to 1970 mainly negative
trends are shown.

## 11  4. Comparison between all-sky and clear-sky SSR records variation

We have used the 59-year data set (1954-2012) in order to quantify the factors controlling the SSR
variations in Athens, Greece, focusing mainly on two known dominant factors, clouds and aerosol
load.

### 15  4.1 The role of clouds

Figure 7 shows the 1954-2012 time series of yearly anomalies based on daily SSR, together with
yearly total cloud coverage in weighted octas. The yearly de-seasonalised SSR values for all-sky
conditions show a drop of ~14% from 1960 to 1970 and then a continuous increase excluding the
Pinatubo period in the early 1990's. Most pronounced positive changes can be seen during the last
15 years with a change of the order of about 15%.

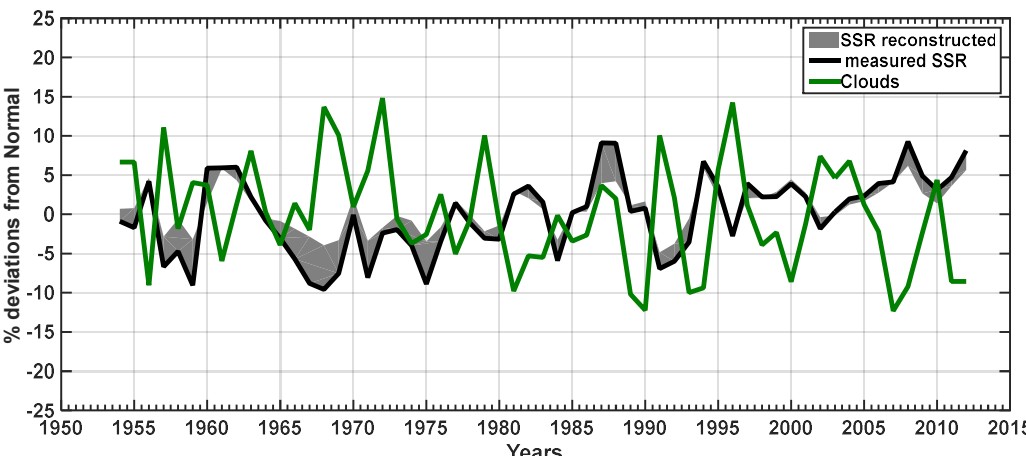

**Figure 7.** De-seasonalised yearly percent deviations from mean for SSR (black line) and cloud
octas. Grey lines are related with measurements possible uncertainties/drifts.
Going back to the measurement uncertainties for the 1954-1983 period where a number of
instruments have been used in order to build the presented time series; we have tried to investigate
possible instrument drifts and their effect on the calculated long term trends. In order to indirectly
try to tackle this issue we included in figure 7 a shaded area representing a possible (one direction)
"uncertainty" based on reconstructing the 1954-1983 series using: the 1984-2012 measured SSR
data and the sunshine duration data for 1954-1983. The reconstruction has been performed in the
same way as the 1900-1953 one. The one direction "uncertainty", points out possible drifts and
instrument exchange related uncertainties. However, that does not mean that we believe more on the
reconstructed through sunshine duration 1954-1983 series than the actual SSR measurements. If this
was the case, we would have decided to present a 1983-2012 high quality measuring period and a
1900-1983 reconstructed one. There are various of such papers published quite recently (small
measuring period compared with the reconstructed one: Garcia et al.,, 2014; [1992-2013
measurements reconstructed back to 1933] and Anton et al., 2017 [1887-1950 using radiative
transfer modelling]) while in our case we would like to try use the best way possible the historical
SSR measurements of NOA during the 1954-83 period.
Using the 1984-2012 measurements and the 1900-1983 reconstruction data set we have
recalculated all trends presented in figure 4 and table 2. Differences for the 15 year window
differences on the calculated trends outside the 54-83 period are less than 1%, with maximum
differences at the late 60's 1-3%. For the 30 year window maximum differences are in the order of
1-2%, while for the 40 year window, maximum differences are less than 1%.
This particular exercise cannot be defined as an uncertainty assessment on the 1954-83
measurements, as reconstructed data cannot be used as a reference. Moreover, SSR is much more
sensitive than SD to aerosol optical depth change. So, in locations where the number of cloudless
days is relatively high SD reconstruction tends to "smooth" the SSR variability, however the
opposite can be said in cases with constant cloudiness.
Figure 8 shows the correlation between annual mean SSR and cloud cover. From the best-fit linear
regression line it is deduced that a -1.54 $MJ\,m^{-2}$ (or -9.6%) change in mean daily insolation
accounts for a change of 1 octa in cloud cover.

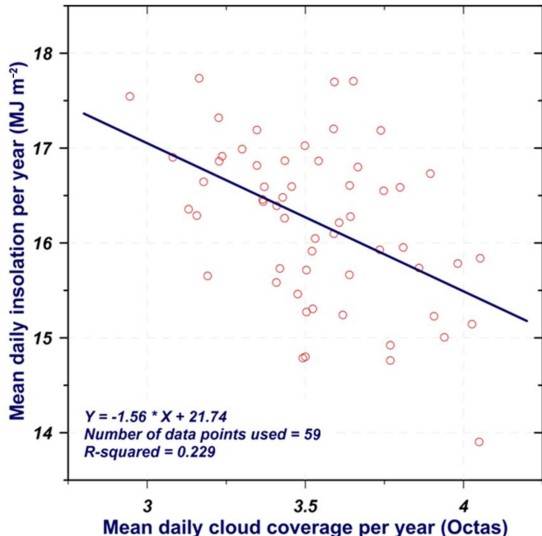

**Figure 8.** Correlation between annual means of daily insolation and cloud cover over the period
1954-2012. The straight line represents the best-fit regression line to the data points. The year 1953
has not been included in the analysis since it does not include measurements for all months.
However, the great scatter of the data points and the low correlation of the two parameters in Figure
8 ($R^2$=0.229) indicate that the cloud cover can only partly explain the changes in SSR. In addition,
there is no significant change in cloudiness over the 59 year period for Athens, Greece. Calculating
linear changes of cloudiness from data shown Figure 8, shows a non-significant change of -0.4%
per decade which can practically have a limited effect on SSR changes during the examined period.
Nevertheless, it is worth mentioning that different cloud properties like cloud optical thickness and
cloud phase, not described by the measurements of cloud cover, can influence SSR.
**4.2 Clear sky records**
In order to minimize the cloud influence and investigate the possible role of direct aerosol effects on
Athens SSR series, we had to select clear-sky (or cloudless) days. We have used daily SSR
measurements from 1954 to 2012 and we have separated the cloudless days according to the criteria
mentioned in Section 2.2.
For considering the SSR seasonality, we have calculated a five-degree polynomial derived from the
maximum daily SSR (for all years of the data set), as a function of the day of the year (Figure 9).
Afterwards we have calculated the ratio of the daily SSR to the SSR calculated by this function.
Seasonal and yearly means of this ratio have been estimated and have been used to describe
cloudless-sky SSR percentage changes on a seasonal and yearly basis. This approach has been
chosen since averaging a random set of cloudless days, within each month during the 59-year
period, could cause solar elevation-related (due to the change of maximum solar elevation within
each month) discrepancies, when calculating the monthly average SSR. It can be emphasized that
the clear sky selection criterion could possibly eliminate a few cases with very high aerosol optical
depth.

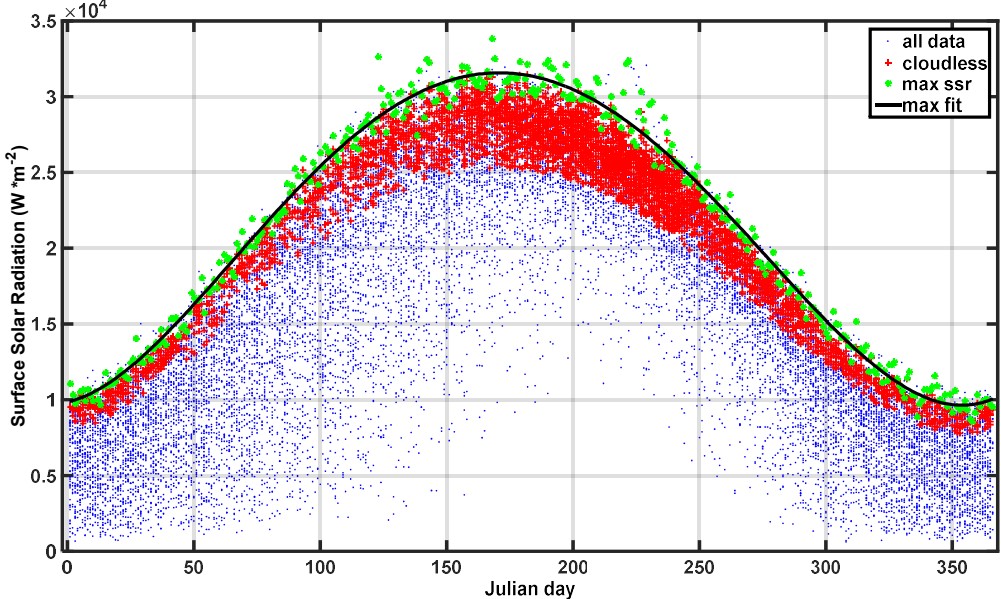

**Figure 9.** Clear-sky SSR measurements (red dots) and all-sky SSR measurements (blue dots)
derived with the cloud octa (cloudiness<1.5) and sunshine duration (SD>0.9) related criteria. The
black line represents the polynomial fit to the daily $SSR_{max}$ values.
Using the clear sky conditions seasonal and yearly averages of SSR have been calculated. The use
of seasonal instead of monthly SSR has been introduced in order to improve the averaging SSR-
related statistics, since the average number of cloudless days (per year) can be relatively low
especially during the winter months. For all cases the ratios of the mean daily cloudless SSR to the
$SSR_{max}$ derived from the daily best-fit curve in Figure 9 has been calculated and deviations of this
ratio from its 59-yr mean have been calculated for each year.

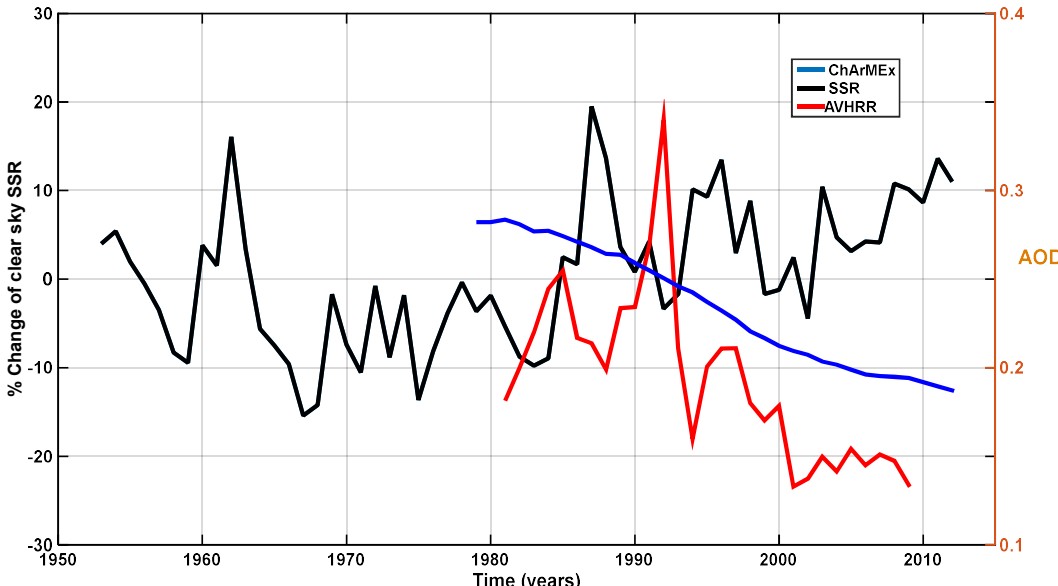

**Figure 10**. Changes in yearly mean SSR to relative to the 1954-2012 average for cloudless sky (in %; black), AVHRR AOD series (red) and ChArMEx AOD climatology (blue; Nabat et al., 2013) for Athens area is shown in the right axis.

Figure 10 shows that most of the SSR variation observed for the measuring period has to be explained by other factors than changes in cloudiness (see figure 7 for variations due to cloudiness). Different seasons with the exception of wintertime show similar patterns to the year-to-year variability. Individual seasonal calculated SSR variability do not exceed by more than ±5%, the SSR variability of all sky data, with the exception of the winter season. Comparing clear sky and all sky yearly mean SSR, we find a high correlation ($R^2$ = 0.71), which can be explained as a combination of: aerosol changes driving the SSR changes and by the number of clear sky days during the year. There is a decrease of more than 15% in the clear sky SSR from the start of the series to the end of 1960's. A decline after 1983 could possibly be related with El Chichon volcanic eruption.

All sky SSR measurements and AOD from AVHRR have been used in order to find the AOD effect in all sky data. For yearly AOD and SSR averages from 1981 to 2009 we have calculated a correlation coefficient of -0.55 with a rate of SSR reduction per 0.1 units of AOD equal with -3.8%. For monthly based comparisons, all months revealed a correlation coefficient of -0.2 with a rate of -1.5% per 0.1 AOD with better results for summer and Autumn months (-0.30, -2.2%/0.1 AOD and -0.30, -1.5%/0.1 AOD)

The Pinatubo-related drop of 6% from the early 1990's to the mid-1993 can also be seen in both cloudless and all-sky datasets and also to the increase in AOD in the AVHRR dataset (Figure 10).

Since, the ~6% drop from 1990 to 1991-1993 is shown for all seasons, we can argue that it
describes the effect of the eruption on SSR data for the Athens station. However, as shown in figure
7 cloudiness for 1991 is also high, while is much lower for 1992 and 1993. Combined with the
stratospheric AOD figure, it seems that 1991 related decrease is also related with cloud increase
while 92 and 93 one with the Pinatubo related aerosol effect.
Concerning the stratospheric AOD in Athens the ChArMEx AOD dataset revealed two main peaks
of 0.12 for 1983 and 0.09 for 1992 due to El Chichon and Pinatubo eruptions, respectively, while
stratospheric AOD after 1995 is lower than 0.01. These two peaks are possibly associated with
decreases in SSR as measured at ASNOA.
Finally, the ~13% change from 1995 to 2012 shown for all skies (Fig. 7) and clear skies (Fig. 10) is
accompanied with a drop of ~25% in AOD measured by AVHRR. The year to year variations of
clear sky SSR series and the AVHRR-related AOD show an anti-correlation with $R=-0.78$ (N=29),
verifying the hypothesis that SSR clear sky changes are associated with aerosol load changes, at
least within the common AVHRR/measurement period (1982-2009).
Similar to the AVHRR data the ChArMEx 4-D aerosol climatology is shown in figure 10, providing
similar conclusions, namely the AOD negative trend of 0.03 or 14% per decade from 1979 to 2012.
Differences between the AVHRR and ChaArMEx data can be explained in part by the different
AOD wavelengths presented here (630 vs 550 nm) and also by a general negative bias of AVHRR
over the Mediterranean compared to AERONET (Nabat et al., 2014). The smooth decline in the
ChArMEx AOD data is due to the method used to build this product and uses the trend and not the
interannual variability which is not included in the global model that was used.

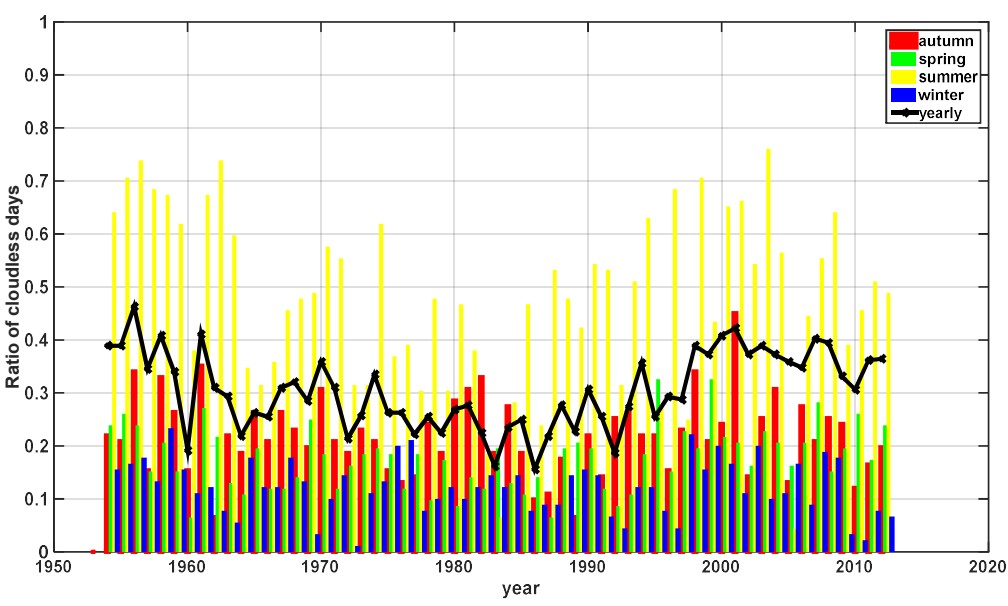

23              Figure 11. Ratio of cloudless vs all days, per season and yearly

In addition to figure 10 we have included figure 11 showing the ratio of cloudless days to all available days for each season and for each year. Figure 11 shows a minimum (less than 30% during a year) of the number of cloudless days from mid-1970's to early 1990's. It is mostly linked

with the decrease of cloudless days during summer months. The figure provides a hint on the SSR relative changes observed during this period, but it cannot directly interpret year to year SSR changes as they depend also on cloud fraction and properties for cloudy days. In addition, it can only partly be linked with fig. 10 as aerosol effects on cloudless sky calculated SSRs depend mostly on AOD levels and not on the number of days included in the calculations.

Differences in the ratio of cloudless days shown in figure 12 and in the almost constant cloud octa variability shown in figure 7 is partly attributed to the different definition of a cloudless day that is based on the cloud radiative effect for fig. 12 and on observation of cloud percentage in the sky for fig. 7. However, this can also be an indication of changes in cloud properties (e.g. change in optically thin clouds that could have small radiation effect but are marked as cloudy conditions from the observer).

In Table 3 we have calculated the linear trends for the 1953-2012 period and for both clear sky and all sky measurements and the 1953-1982 and 1983-2012 sub-periods for clear sky measurements. Results shows comparable changes per decade (2% for the clear sky and 1.5% for the all sky cases). Seasonal analysis show that clear sky trends for summer, autumn and winter months are higher than the ones derived for all skies. Such differences are linked with the seasonal variability and long-term changes in cloudiness for the specific seasons.

Table 3: Clear sky and all sky data trends comparison for the whole 1953-2012 period and the two 30-yr sub-periods (% per decade). Percentages in parenthesis show the limits of the 95% confidence bounds.

| Season | Clear sky 1953-2012 | All skies 1953-2012 | Clear sky 1953-1982 | Clear sky 1983-2012 |
|---|---|---|---|---|
| Winter | 0.91 (±2.31) | -6.43 (±3.83) | -7.01 (±3.16) | 0.55 (±2.41) |
| Spring | 1.22 (±1.12) | -0.60 (±3.10) | -0.92 (±1.11) | 2.62 (±1.97) |
| Summer | 2.03 (±0.78) | -1.14 (±2.90) | -0.36(±0.83) | 1.31(±0.81) |
| Autumn | 2.74 (±1.37) | -1.28 (±3.42) | -1.03(±1.84) | -1.48 (±1.73) |
| Year | **2.17 (±1.21)** | **-2.33** (±2.28) | -1.44(±2.35) | 1.94 (±2.08) |

Clear sky results for the 1953-2012 period show significant positive changes in SSR for all seasons except winter. Looking individually at the 1953-1982 and 1983–2012 periods we have calculated significant negative trends only for the winter over the first and for summer and spring over the second.

The effect of various parameters on SSR has been discussed by Kambezidis et al. (2016) in their study about the global dimming/brightening effect over the Mediterranean in the period 1979-2012. They show that the influence of parameters related to the atmospheric transparency, like water vapor, aerosols and trace gases, as well as changes in the surface albedo on SSR have been larger in the southern parts of the Mediterranean, over the Balkan countries and central Turkey. This outcome is in agreement with the conclusion of the present study that other factors than cloudiness play significant role in the SSR variations.

A comparison of the SSR results in Athens with visibility observations since 1931 (Founda et al., 2016) did not show any correlation among SSR and horizontal visibility. For the first part of the common dataset (1930-1959) the visibility decline is accompanied with a SSR increase. However from 1950 till today visibility shows a monotonical decrease. The steep visibility decrease from 1931 till early 90's is not accompanied by a relative SSR decrease excluding individual sub-periods.

However, simulated SSR is driven purely by changes in sunshine duration, in this case the SD variability in founda et al., 2014 is almost stable after 1950 so SD can not be also linked with the visibility reported decrease. Studying the literature for similar cases, similar conclusions have been drawn by Liepert and Kukla (1997) showing an SSR decrease over 30 years of measurements accompanied by a visibility increase and no significant changes in the cloud cover conditions, in Germany. This Athens SSR vs visibility relationship can be partly explained by the fact that SSR and visibility have different response on cloud conditions, water vapor and rainfall, and also by the fact that visibility is affected by aerosols only in the first few hundred meters above the surface, while SSR is affected by the columnar AOD, which in the case of Athens can be significantly different due to aerosol long-range transport in altitude (e.g. Saharan dust; Léon et al., 1999).

## 5 Conclusions

Surface solar radiation (SSR) at National Observatory of Athens, in the center of the city, is presented using a unique dataset covering a period of 59 years (1954-2012). Sunshine duration (SD) records for another 54 years have been used as a proxy to reconstruct SSR time series for the period from 1900 to 2012.

The data accuracy of such historic radiation dataset is more difficult to be assessed especially going back to the 50' and 60's where instruments, operational procedures and quality control were not at the same level as in the recent 30 years. Quality assessment procedures in the presented time series have been applied with criteria based on instrument characteristics and the availability of additional collocated measurements. Year to year fluctuations of the measured SSR in addition to the reversal of the downward tendencies at the ASNOA site adds credibility to the measured variations. That is because a typical radiometer behavior is to lose sensitivity with time indicating spurious downward, but not upward trends. The more recent (after 1986) SSR measurements can be characterized as high-quality radiation data with known accuracy. Considering the measurements from 1954 to 1970 there has been sporadic reports mentioning the homogenization and calibration procedures, while for 1970 to 1986 there is more information on the instrument quality control.

Reporting of the results from the 1954-1986 period should be accompanied with the fact that the uncertainties of the measurements of this period are linked with higher uncertainties than after 1986. For the reconstruction of the 1900-1953 series, only the 1983-2012 SSR and SD measurements were used in order not to link possible instrument uncertainties to the extrapolated period. However, reconstruction of the 1900-1983 time series using the 1984-2012 dataset leads to small differences in the determination of the long term trends, especially for more than 20 year running average windows.

De-seasonalized SSR data analysis from 1900 to 2012 showed high month to month variability that could reach up to 25%, mainly related with monthly cloudiness variations. During the period 1910-mid-1930s where only few datasets have reported worldwide SSR results, we observe a -2.9% per decade or a total of -8.7 % decrease in SSR, assuming linear changes in SSR during this period. This early dimming was followed by a +5% per decade increase from 1930 to the 1950s. Similar results have been found at Washington DC and at Potsdam, Germany (Stanhill and Achiman, 2016).

They have reported an early brightening at both locations in the 1930's. For the SSR measurement period of 1953 to 1980, European related studies presented in Wild (2009) showed a -1% down to -7% change per decade in SSR measurements over various European sites (dimming period). For the Mediterranean region, Manara et al. (2016) showed a decrease of the order of -2% to -4% per decade in Italy. We are reporting a change in SSR of -2% per decade in Athens. Finally, for the brightening (1990-2012) phase again Wild et al. (2009) reported a +1.6% up to +4.7% per decade positive change in SSR while we have calculated a +1.5% per decade, which is lower than the +(3-6)% per decade reported in Manara et al. (2016) for Italy. A summary of the above findings can be seen in table 4.

Table 4: Summary of per cent SSR changes per decade for various locations

| Period | Location | Trend % per decade | Reference |
|--------|----------|----------|-----------|
| 1893-2012 | Potsdam, Germany | 0.71 | Stanhill and Achiman, 2014 |
| 1900-2012 | Athens, Greece | 0.40 (±0.26) | This work |
| 1959-1988 | Europe | –2.0 | Ohmura and Lang, 1989 |
| 1971-1986 | Europe | –2.3 | Norris and Wild, 2007 |
| 1959-1985 | Italy | –6.4(±1.1) / -4.4(±0.8) | Manara et al, 2016 |
| 1953-1982 | Athens, Greece | -2.33(±2.28) | This work |
| 1985-2005 | Europe | 2.5 | Wild, 2009 |
| 1990-2012 | Italy | 6.0 (±1.1) / 7.7 (±1.1) | Manara et al, 2016 |
| 1986-2013 | Athens, Greece | 0.80 (±1.96) | This work |

The decadal variations of SSR measured since 1954 at Athens, Greece, originate from the
alterations in the atmosphere's transparency (namely by clouds and aerosols). Using an analysis of
SSR calculations of all sky and clear sky (cloudless) days we end up that since cloud cover changes
during the 59 period were very small, most of the observed decadal changes can be related with
changes in the aerosol load of the area. An additional hint in support of this conclusion is the high
correlation of clear sky and all sky yearly SSR. We also found an anti-correlation between either
clear sky and all sky SSR measurements and AOD time series from AVHRR (1981-2009) or
ChArMEx (1979-2012). Looking at linear trends over the 59 year period, clear sky changes per
decade were 2% while it was 1.5% for all sky conditions. The most pronounced changes have been
calculated for summer and autumn seasons (2% and 2.7% respectively).
**Acknowledgements**
The authors wish to thank all the past and present NOA staff members who carefully collected and archived the long-
term data used in this study. This study contributes to the Chemistry-Aerosol Mediterranean Experiment (ChArMEx)
Work package 6 on trends. The work was partly funded by the Greek national project "Aristotelis", work package 1:
"Study of long term variations of Solar Radiation in the region of Athens". A. S. L. was supported by postdoctorial
fellowships (JCI-2012-12508 and RYC-2016-20784) and a project (CGL2014-55976-R) funded by the Spanish
Ministry of Economy, Industry and Competitiveness. We would like to thank the anonymous reviewers, Dr. Tanaka and
the editor Dr. Dulac for their efforts on substantially improving this work.

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
