# Peer review of "Long-term series of surface solar radiation at Athens, Greece"

_Atmospheric Chemistry and Physics, 2017_

## Referee Comment (RC1) · KT Tanaka (Referee) · 10 Apr 2017

This study presented all-sky SSR data observed in Athens from 1952 to 2012 and extended the data back to 1900 based on a statistical relationship with SD data. They also derived clear-sky SSR data by removing data on cloudy days from the all-sky SSR data. They documented the data sources in detail and inspected the trends and variabilities of the SSR data in separate time periods and with different time resolutions (daily, monthly, and annually). Among others, the paper particularly explored the question as to what are the sources of decadal variabilities in SSR data in Athens: aerosols or clouds. They compared the SSR data with cloud cover and AOD data and concluded that the SSR decadal variabilities were caused mostly by aerosols rather than clouds.

The significance of this paper may be limited because they deal with the SSR data only

at one location (Athens) and support/confirm previous results drawn from analyses dealing with wider regions (countries and continents) as evident in Table 4 (also, for example, dominant role of aerosols in determining the decadal SSR trend in Europe (Norris and Wild 2007; Ruckstuhl et al. 2008; Wang et al. 2012)). On the other hand, the long-term SSR data including those reconstructed and the accompanied analysis does make a contribution to the debate associated with global dimming and brightening (Wild 2009).

Furthermore, the paper discusses relevant previous studies in Sections 1 and 4, but it also needs to state more explicitly what are scientifically new in this study. In terms of the presentation, I think that the text could generally be condensed a lot by organizing the argument. I spotted a number of errors in writing and pointed out several of them as minor comments.

As a main point, I argue that, while the authors looked into the SSR data from several angles to arrive at the abovementioned conclusion, they failed to show a straightforward analysis looking directly into the correlation between the all-sky SSR and the AOD. As far as I understand, the evidences they provide in the current manuscript are essentially a combination of high correlations between the clear-sky SSR and the AOD and also between the clear-sky SSR and the all-sky SSR (Figure 11 and text). Also, as a counter evidence, they indicated a weak correlation between the all-sky SSR and the cloud octa (Figures 8 and 9). These evidences all point toward the dominant role of aerosols in determining the all-sky SSR, but these are all indirect evidences. I think they could make the same point more clearly by directly presenting the relationship between the all-sky SSR and the AOD.

I have minor comments as follows:

- The abstract states that a decrease of 2.9%/decade in SSR from 1910 to 1940. I wonder why the trend from this particular period is selectively highlighted within the extrapolated period of 1900-1952. Table 2 indicates a small increase of 0.04%/decade

from 1900 to 1952, which is clearly different from above.

- Also in the abstract, I could not find where the difference of 4.5% comes from. Table 2 indicated approximately 3.1% but for a slightly different period. The winter period shows the largest change, unlike what is stated in the abstract.

- Page 2, Line 21: Regarding the discussion on SSR changes in polluted and pristine areas, I believe that this is still an issue of controversy but two recent studies (Imamovic et al. 2016; Tanaka et al. 2016) showed otherwise, which can be reflected to this statement.

- Page 3, Line 2: Fix the citation style.

- Page 3, Line 6: Remove "explain".

- Page 3, Line 22: Figure 1 of (Ohmura 2009) also makes a clear case for this statement.

- Page 4: Somewhere in the text (not necessarily in this page), the discussion could touch on aerosol-clouds interactions to acknowledge that the two factors (aerosols and clouds) are not completely mutually exclusive in explaining SSR trends.

- Page 11, Line 14: I am trying to speculate what causes the weak correlation in summer. The paper cites small ranges of variables in summer as a reason for weak correlation, but how exactly do the range affect R2 values? Later in the paper (Figure 12), the number of cloudless days in summer is generally large, compared in other seasons. Could the number of cloudless days influence the correlation level?

- Page 12, Line 10: "light grey" should be "light blue" from what I can see from the figure.

- Page 13, Line 1: Separate "late1930's" into two words.

- Page 13, Line 7: It may be useful to break up the 1900-1952 period into two because the text discusses the trend till late 1930s and the trend that follows separately.

- Page 13, Line 11: Remove comma after 2012.

- Page 14, Lines 10-13: It needs to be specific which region it refers to. The trend of global anthropogenic BC emissions during 1910-1950 does not decline but rather levels off (Lamarque et al. 2010).

- Page 15, Line 7: Remove comma after Figure 6.

- Page 16, Line 3: Is the left panel of Figure 7 essentially same with Figure 5? If so, the left panel does not have to be shown as it is redundant.

- Page 18, Line 9: "non significant" needs to be connected by hyphen.

- Page 20, Line 11: the SSR line should be "black" rather than "blue".

- Page 22, Line 7: Would there be any possible explanation why only the clear-sky SSR trend in winter is negative? A similar result was obtained for the all-sky SSR (Table 2).

- Page 22, Line 25: The discussion on visibility can be part of the discussion, not the conclusion. Visibility has not been brought up since the introduction.

- Page 23, Line 3: "drown" should be "drawn".

References

Imamovic A, Tanaka K, Folini D, Wild M (2016) Global dimming and urbanization: did stronger negative SSR trends collocate with regions of population growth? Atmospheric Chemistry and Physics 16 (5):2719-2725. doi:10.5194/acp-16-2719-2016

Lamarque JF, Bond TC, Eyring V, Granier C, Heil A, Klimont Z, Lee D, Liousse C, Mieville A, Owen B, Schultz MG, Shindell D, Smith SJ, Stehfest E, Van Aardenne J, Cooper OR, Kainuma M, Mahowald N, McConnell JR, Naik V, Riahi K, van Vuuren DP (2010) Historical (1850–2000) gridded anthropogenic and biomass burning emissions of reactive gases and aerosols: methodology and application. Atmos Chem Phys 10 (15):7017-7039. doi:10.5194/acp-10-7017-2010

[Figure]

Norris JR, Wild M (2007) Trends in aerosol radiative effects over Europe inferred from observed cloud cover, solar "dimming," and solar "brightening". Journal of Geophysical Research: Atmospheres 112 (D8):D08214. doi:10.1029/2006JD007794

Ohmura A (2009) Observed decadal variations in surface solar radiation and their causes. Journal of Geophysical Research 114:D00D05. doi:10.1029/2008jd011290

Ruckstuhl C, Philipona R, Behrens K, Collaud Coen M, Dürr B, Heimo A, Mätzler C, Nyeki S, Ohmura A, Vuilleumier L, Weller M, Wehrli C, Zelenka A (2008) Aerosol and cloud effects on solar brightening and the recent rapid warming. Geophys Res Lett 35 (12):L12708. doi:10.1029/2008gl034228

Tanaka K, Ohmura A, Folini D, Wild M, Ohkawara N (2016) Is global dimming and brightening in Japan limited to urban areas? Atmos Chem Phys Discuss 2016:1-50. doi:10.5194/acp-2016-559

Wang KC, Dickinson RE, Wild M, Liang S (2012) Atmospheric impacts on climatic variability of surface incident solar radiation. Atmos Chem Phys 12 (20):9581-9592. doi:10.5194/acp-12-9581-2012

Wild M (2009) Global dimming and brightening: A review. J Geophys Res 114:D00D16. doi:10.1029/2008jd011470
* * *

---

## Author Comment (AC1) · 14 Nov 2017

We would like to thank Dr. Tanaka for his fruitful comments.

**ALL sky vs AOD**

All sky SSR measurements and AOD from AVHRR have been used in order to find the AOD effect in all sky data. For yearly AOD and SSR averages from 1981 to 2009 a correlation coefficient of -0.55 was calculated with a rate of SSR reduction per 0.1 units of AOD equal with -3.8%.For monthly based comparisons, all months revealed a correlation coefficient of -0.2 with a rate of -1.5% /0.1 AOD with better results for summer and autumn months (-0.30, -2.2%/0.1 AOD and -0.30, -1.5%/0.1 AOD)

[Figure]

*- The abstract states that a decrease of 2.9%/decade in SSR from 1910 to 1940. I wonder why the trend from this particular period is selectively highlighted within the extrapolated period of 1900-1952. Table 2 indicates a small increase of 0.04%/decade from 1900 to 1952, which is clearly different from above.*

We have altered the abstract including:
Very small (0.02%) changes in SSR from 1900 to 1952, including a maximum decrease of 2.9% per decade in SSR from when taking in to account the 1910 to 1940 period, assuming a linear change in SSR.

*- Also in the abstract, I could not find where the difference of 4.5% comes from. Table 2 indicated approximately 3.1% but for a slightly different period. The winter period shows the largest change, unlike what is stated in the abstract.*

The 4.5% comes from figure 6 as an average of the difference of the 12 months.
Adding the trends of 1953-82 and 1983-2012 gives a slightly different result because mathematically these two individual percentages are calculated using de-seasonalized data using different (for the two periods) mean month values.

*- Page 2, Line 21: Regarding the discussion on SSR changes in polluted and pristine areas, I believe that this is still an issue of controversy but two recent studies (Imamovic et al. 2016; Tanaka et al. 2016) showed otherwise, which can be reflected to this statement.*

Reference to these studies and corresponding discussion has been added.

*– Page 3, Line 2: Fix the citation style.*

Style has been fixed

*- Page 3, Line 6: Remove "explain".*

Extra word has been removed.

*- Page 3, Line 22: Figure 1 of (Ohmura 2009) also makes a clear case for this statement.*

Reference to this work has been added.

*- Page 4: Somewhere in the text (not necessarily in this page), the discussion could touch on aerosol-clouds interactions to acknowledge that the two factors (aerosols and clouds) are not completely mutually exclusive in explaining SSR trends.*

Added sentence in page 2:"However, due to the aerosol-cloud interactions and the

aerosol indirect effect on SSR (e.g., Rosenfeld et al., 2014), the two factors (clouds

and aerosols) are not completely mutually exclusive in explaining SSR changes."

*- Page 11, Line 14: I am trying to speculate what causes the weak correlation in summer. The paper cites small ranges of variables in summer as a reason for weak correlation, but how exactly do the range affect R2 values? Later in the paper (Figure 12), the number of cloudless days in summer is generally large, compared in other seasons. Could the number of cloudless days influence the correlation level?*

The weak correlation is probably caused by the very low variability of the SDu/SDmax and the SSR/SSRmax ratios. Below an example for July and August correlations where in XX' axis is the SDu/SDmax and in the YY' axis is the SSR/SSRmax. Large number of cloudless days in the summer is exactly the reason for this low variability. So the calculated Ångström factors for monthly based analysis, based on this example can not be used as only a 12.6% and 8% of the variability of the reconstructed (1900-1953) Julys and Augusts could be explained using these method.

[Figure]

In the initial submission we have used the Ångström related formula in order to calculate SSR and SD related functions. This method includes the theoretical SSR and SD maximum values that insert an uncertainty for such calculations. After the reviewer's comment we decided to replace this method with the one used by Sanchez-Lorenzo and Wild (2012). One additional reason to test this method (as mentioned also in the paper) was the fact that monthly based calculated SD to SSR conversion functions had high uncertainty, linked with the very small SD/SDmax absolute variability especially for summer months.

In this new approach (Sanchez-Lorenzo and Wild, 2012) we did not use SSR and SD theoretical maxima in order to normalize the two factors, but monthly anomalies of SSR and SD have been used for a common measuring period and then the monthly coefficients of the regression of SSR and SD anomalies were used in order to reconstruct the 1900-1953 time series. The regression statistics of these monthly based SSR and SD anomalies analysis showed much better results from the Ångström method. As an example (and included in the new manuscript) statistics and graphs are shown below.

[Figure]

| month | Jan | Feb | Mar | Apr | May | Jun | Jul | Aug | Sep | Oct | Nov | Dec |
|-------|-----|-----|-----|-----|-----|-----|-----|-----|-----|-----|-----|-----|
| a | 22.47 | 34.99 | 39.53 | 46.65 | 57.88 | 44.67 | 51.87 | 46.23 | 34.28 | 28.02 | 27.32 | 23.10 |
| R | 0.842 | 0.895 | 0.887 | 0.840 | 0.799 | 0.757 | 0.773 | 0.572 | 0.812 | 0.888 | 0.916 | 0.842 |

The correlation coefficients show that SDU can explain from 65% to 82% of the variability of the SSR monthly anomalies. This additional verification analysis shows that the method used in this work is in accordance with important already published results. (e.g. Sanchez-Lorenzo and Wild, 2012) that have been analysed 17 stations with very long term SDU series.

After having calculated the reconstructed series with this method we have compared the yearly and monthly SSR deviations with the ones calculated with the Angstrom method using the yearly functions (initial submission). The results in yearly basis for all 1900-1953 period differ at a maximum by 1%.

The agreement of these two results shows that in the case that SD measurements in the past have no particular quality issues, then SSR can be reconstructed with the 65-82% explained variability already mentioned.

Finally we have decided to keep the new method on the revised document and include the (yearly based) Ångström results as a verification. The inclusion of this method had a direct impact on all related figures 2, 3, 4, 5, 8 and tables describing trends that include the 1900-1953 period. As already reported the differences were small but still all the plots and tables have been replaced with the new ones calculated based on the Sanchez-Lorenzo and Wild (2012) method.

*- Page 12, Line 10: "light grey" should be "light blue" from what I can see from the figure.*

The colour has been described properly.

*- Page 13, Line 1: Separate "late1930's" into two words.*

Suggested change has been edited.

*- Page 13, Line 7: It may be useful to break up the 1900-1952 period into two because the text discusses the trend till late 1930s and the trend that follows separately.*

We think that the current period break up into 1900-1952-1983-2012 periods is already a bit of a mix up for the reader. The basic idea behind this was that the 1900-1952 period is simulated SSR and the 1983-2012 two times 30 year measurement periods that could be also compared with each other. Figures like 5 and 7 could be used to retrieve any SSR % change for any time window and there can be readers that could be interested in a very specific period during these 112 years of reconstructed & measured SSRs.

*- Page 13, Line 11: Remove comma after 2012.*

Comma has been removed.

*- Page 14, Lines 10-13: It needs to be specific which region it refers to. The trend of global anthropogenic BC emissions during 1910-1950 does not decline but rather levels off (Lamarque et al. 2010)*

[Figure]

**Fig. 4.** Total annual emissions (anthropogenic, shipping and biomass burning) of NO$_x$ (Tg(N)/year) for 1850 (top left), 1900 (top r, 1950 (bottom left) and 2000 (bottom right).

The sentence has been changed to:
"Nevertheless, early dimming and brightening periods have been reported in Stanhill and Achiman (2016). The results can be partly supported by trends in anthropogenic black carbon (McConnell et al., 2007; Lamarque et al., 2010) and biomass-burning (Lamarque et al., 2010) emissions in Europe."

*- Page 15, Line 7: Remove comma after Figure*

Comma has been removed.

*6. - Page 16, Line 3: Is the left panel of Figure 7 essentially same with Figure 5? If so, the left panel does not have to be shown as it is redundant.*

The left panel in figure 7 is linked with figure 5 as it represents a sub period. Mathematically it is not the same as part of figure 5 as de-seasonalized monthly and yearly SSRs have been calculated only for the provided (fig. 7) and not total (fig. 5) period. But we agree with the reviewer that essentially they are the same so we deleted figure 7a but included the discussion on this figure on the existing paragraph.

*- Page 18, Line 9: "non significant" needs to be connected by hyphen*

Hyphen has been added.

*- Page 20, Line 11: the SSR line should be "black" rather than "blue".*

The colour has been described properly.

*- Page 22, Line 7: Would there be any possible explanation why only the clear-sky SSR trend in winter is negative? A similar result was obtained for the all-sky SSR (Table 2).*

There is no straight forward explanation for this negative winter trend. There are various aspects related with the seasonal trend calculation for wintertime such as:
- Wintertime clear sky statistics include more uncertainty due to the more frequent presence of clouds and the fewer clear sky points available.
- Clear sky changes and trends are linked with aerosol changes. For Athens area absolute AOD values for winter are minimum compared with other seasons.

*- Page 22, Line 25: The discussion on visibility can be part of the discussion, not the conclusion. Visibility has not been brought up since the introduction.*

The section has been transferred to the discussion section and only the conclusions of visibility related discussion has been left to the conclusion section.

- Page 23, Line 3: "drown" should be "drawn".

*Typo has been corrected*

**References**

Imamovic A, Tanaka K, Folini D, Wild M (2016) Global dimming and urbanization:

did stronger negative SSR trends collocate with regions of population growth? Atmospheric

Chemistry and Physics 16 (5):2719-2725. doi:10.5194/acp-16-2719-2016

Lamarque, J.-F., Bond, T. C., Eyring, V., Granier, C., Heil, A., Klimont, Z., Lee, D., Liousse, C., Mieville, A., Owen, B., Schultz, M. G., Shindell, D., Smith, S. J., Stehfest, E., Van Aardenne, J., Cooper, O. R., Kainuma, M., Mahowald, N., Mc-Connell, J. R., Naik, V., Riahi, K., and van Vuuren, D. P.: Historical (1850–2000) gridded anthropogenic and biomass burning emissions of reactive gases and aerosols: methodology and application, Atmos. Chem. Phys., 10, 7017–7039, doi:10.5194/acp-10-7017-2010, 2010.

Ohmura, A.: Observed decadal variations in surface solar radiation and their causes, J. Geophys. Res., 114, D00D05, doi:10.1029/2008JD011290, 2009.

Sanchez-Lorenzo, A., and Wild, M.: Decadal variations in estimated surface solar radiation over Switzerland since the late 19th century, Atmospheric Chemistry and Physics 12.18: 8635-8644, 2012.

Stanhill, G. and Achiman, O.: Early global radiation measurements: a review. Int. J. Climatol.. doi:10.1002/joc.4826, 2016

Tanaka K, Ohmura A, Folini D, Wild M, Ohkawara N (2016) Is global dimming and

brightening in Japan limited to urban areas? AtmosChem Phys Discuss 2016:1-50.

doi:10.5194/acp-2016-559

McConnell, J. R., Aristarain, A., Banta, J., Edwards, P., and Simoes,J.: 20th Century doubling in dust archived in an AntarcticPenisula ice core parallels climate change and desertification inSouth America, P. Natl. Acad. Sci., 104, 5743–5748, 2007.

D. Rosenfeld, Sherwood S., Wood, R., Donner, L., Climate Effects of Aerosol-Cloud Interactions, Science, Volume 343, Issue 6169, pp. 379-380, 2014

---

## Author Comment (AC3) · 14 Nov 2017

Thank you.

———————————————————

---

## Author Comment (AC2)

**Referee 3**

We would like to thank the reviewer for his/her comments. We have tried to answer as detailed as possible.

*Page 3, line 4: AOD trend of 0.05 per decade: in what period?*

The period 1979-2009 was added in the text.

*P. 3, l. 9: "This attenuation may be much larger ..."*

This sentence has been restated.

*P. 5, l. 17: "Maximum error on the daily integral SSR..."*

This sentence has been restated

*Table 1 and associated discussion: there are relatively long periods between instrumental changes (up to 6 years). How the radiometers were calibrated prior to 1992? Which was the reference scale? Were the instruments compared with the old one before substitutions? Was the occurrence of instrumental drifts checked?*

All instruments used in the study were accompanied by the calibration certificate of the manufacturer in the radiation scale used at that time. This was primary used for calibrating the instruments. For the presented time series the information on homogenization methods and major corrections made is getting sparser going back in time. For instrument changes till 1968 there are only reports (no actual data) that refer to overlapping periods of measurements for two instruments with just comments on overlapping related corrections and no information on the magnitude of corrections. The two Eppley pyranometers have been calibrated 2 times during the period 1975-1986 in the World Radiation Center in Davos Switzerland and the calibrations acquired have been used for processing the data.

There are several publications that have used the 1954-1983 time series of NOA that are reported in the following comment.

More information on the instrumentation in use and details about temperature correction:

For Solarigraph Gorezynski pyranometers, used during the period 1953-1959, it is well known (Coulson, 1975; Robinson, 1966) that no temperature compensation was provided in these instruments. The temperature coefficient was about 0.0015-0.0020 per $1^{o}C$ in the sense of decreasing sensitivity with increasing temperature. This theoretical rate have been used.

Two Eppley 180 pyranometers have been used in NOA between 1960 and 1973 period. Solar radiation measurements published in the official NOA's Bulletin were temperature compensated improving considerably the performance of these two pyranometers. According to Drummond (1965), accuracies of the order of ±2-3% are attainable for daily summations of radiation with temperature compensated Eppley 180 pyranometers. That uncertainty does not include instrument drifts with time (if any).

All Eppley 8-48, 8-48A and PSP pyranometers, used in NOA for solar radiation measurements from 1974 until now, are equipped with a built-in temperature compensation circuit. According to Coulson (1975) and Hulstrom (1989), Eppley 8-48 pyranometers provide a signal which is independent of temperature to within ±1.5% from -20 to +40$^{o}$C. Also, Eppley PSP pyranometers present a reduced temperature dependency of sensitivity on ambient temperature of ± 1% from -20 to +40$^{o}$C.

On the other hand, the uncompensated Eppley 180 pyranometers are subject to a significant dependence of sensitivity on the temperature of the instrument. The sensitivity decreases with increasing temperature by between 0.05 to 0.15% per 1$^{o}$C rise in the temperature over the range -50 to +40$^{o}$C (Coulson, 1975; Robinson 1966).

*P. 6, l. 6-12: the application of different data selection criteria, with the addition of quality checks based on the diffuse irradiance, may potentially influence the results of the trend analysis, Did the author check that this is not the case?*

The addition of the diffuse irradiance synchronous data in the quality control procedures in the recent years, of course, decreases the uncertainty of the daily, monthly SSR calculations, compared with the one for past years. However, there is no specific scientific evidence or hint in the calculations that points towards a systematic overestimation or underestimation of the SSR data when not using the diffuse radiation "controls". Such an improvement on the quality control methods is in line with most SSR measuring monitoring stations with long term SSR data series published, worldwide.

*P. 7.l. 1-2: as far as I understand, the night-time dark signal was subtracted from daytime measurements. This procedure reduces but does not eliminate the thermal offset of the instruments. It must be taken into account that the different types of radiometers display a quite different thermal offset; in general, this is much larger for PSP than for 8-48 or 180◦ pyranometer. Thus, a systematic overestimate of the SSR in daytime, up to 3-4 W/m2, is possibly present in the data after 1989. This may potentially produce an artificial positive trend in SSR in the recent year; at least, an additional uncertainty should be considered in the trend analysis. Did the authors take into account this effect?*

The effect has been tackled based on the temperature corrections that have been mentioned in the comment above but mostly by the overlapping comparisons and homogenization during the 1986 (last) change of instrumentation. Comprehensive studies (Solar and Infrared Radiation Measurements, Energy and the environment, by Frank Vignola, Joseph Michalsky, Thomas Stoffel, CRC Press, 2016) have pointed out to the PSP related thermal offset issue. In our case the subtraction of the night time dark signal (more specific the mean of the previous and next night signal was subtracted for a specific day) reduces at least in half the problem. However, in order to answer to the reviewer question, this (remaining 1-2 W/m$^{2}$ ?) was not considered in our analysis as another part of this have been tackled through the overlapping measurements/homogenization procedures. The possibility of such an uncertainty is mentioned in the new manuscript.

Concerning the first three comments on the instrument performance for the period 1954-1986 we decided to include a list of publications that have used the SSR presented time series for this period:

- Macris, 1959, have used the 1954-1956 SSR measurements to identify the relationship of SSR and sunshine duration.
- Katsoulis and Papachristopoulos, 1978, have used the NOA SSR data from 1960 to 1976 in order to calculate SSR statistics for daily, seasonal and yearly solar radiation levels for Athens, Greece.
- Notaridou and Lalas, 1979, have been used the 1954-1976 SSR data from NOA in order to verify an empirical formula on global net radiation over Greece.
- Flocas, 1980 have used the 1961-1975 SSR time series to compare them with sunshine duration data for the period.
- Kouremenos et al., 1985 have used the SSR data from 1955-1980 to correlate changes with various atmospheric parameters.
- Zabara, 1986 have used the 1965-1980 time series to verify a developed method that calculated monthly solar radiation.
- Katsoulis and Leontaris, 1981, have used the 1960-1977 data to verify tools describing the solar radiation distribution over Greece.

We have included these references in the new document.

The reference to these studies does not automatically means that the 1954-1986 data do not include uncertainties related with the calibration frequency and quality control, but they are mentioned as a proof of the scientific data quality for the given period, based on the work of various solar radiation related scientists.

*Moreover: there are some rapid changes in the series that may require additional scrutiny; some of these seem to be in correspondence or close to the dates of the radiometers' replacements (e.g., possibly in 1960, 1968, 1973). This seems even more evident in figure 8 from the de-seasonalized monthly mean SSR. Was the presence of step-changes in the series, mainly in corrispondence with instrument replacement, checked?*

Specifically on the changes mentioned and linked with the instrument changes.

1960: We have to rely on the reports that state that the two time series have been homogenized, as we did not find other than the officially published NOA bulletin data, that did not include overlapping measurement periods.
1968: Same as 1960.
1973:The Eppley pyranometer, type 8-48A, was recalibrated in 2004 by the Laboratory of Meteorological Device Calibration (LMDC, Psiloglou, 2016), with an uncertainty of 1.2%. It presents a decrease in sensitivity with a rate of 0.5% per year.

General comment

As mentioned in the uncertainty analysis section, the uncertainty for at least the first two decades of measurements is higher than the one in the last 30 years. According to the log books of instrument users, comparisons of two instruments, before exchanging from the one to the other, has been performed and data have been corrected. As reported we have added a large paragraph describing this aspect and also pointed out in the abstract and conclusion sections this issue (at he end of this comment).

One of the "problems" with the time series is that compared with other European long term SSR studies, the changes of SSR during brightening and dimming periods are visible but to a

lesser extent. This, in addition to the fact that the uncertainty of the 1953-1982 period is higher compared to the last 30-years period leads to the conclusion that the small long term changes at least for the first period (1953-1982) could be within the instrument's uncertainty.

However, small drifts and enhanced uncertainties are linked worldwide with past data of such series, as instrumentation and quality control procedures are improving with time and information quality on the initial decisions is more frequently and scientifically reported.

In order to indirectly try to tackle the question of the data quality of the 1954-1983 period including instrument changes and unaccounted possible instrument drifts, compared with the 1984-2012 period, we investigated the following:

We included in figure 8 a shaded area representing a possible (one direction) "uncertainty" based on reconstructing the 1954-1983 series using: the 1984-2012 measured SSR data and the sunshine duration data for 1954-1983. The reconstruction has been performed in the same way as the 1900-1953 one (new method, see comment below). The one direction "uncertainty", points out possible drifts and instrument exchange related uncertainties. However, that does not mean that we believe more the reconstructed through sunshine duration 1954-1983 series than the actual SSR measurements. If this was the case, we would have decided to present a 1983-2012 high quality measuring period and a 1900-1983 reconstructed one. There are various of such papers published quite recently (small measuring period compared with the reconstructed one: Garcia et al.,, 2014; [1992-2013 measurements reconstructed back to 1933] and Anton et al., 2017 [1887-1950 using radiative transfer modelling]) while in our case we would like to try to use the best way possible the historical SSR measurements of NOA during the 1954-83 period.

[Figure]

Figure 8 including the reconstructed series and one direction uncertainty estimation (shade area)

In order to investigate more on the reviewer question about the link of data quality of the exchanging instruments period to the calculated trends we have calculated:

A reconstructed SSR series for 1954–1983 using the 1984-2012 SSR-SD calculated functions has been performed, exactly the way the extrapolation has been performed for the 1900-1953

period. So this case is similar to as if we had only measurements after 1984 and we extrapolated back to 1900 using the functions calculated from the 1984-2012 period.

Following, we present the trends per decade using the 15, 30 and 40 year windows. Blue curves represent the trends calculated using the measured 1954-1983 SSR data (they are the same as the ones presented in figure 4 of the paper) and orange lines the trends with the reconstructed (1900-1983) data. (figures a: the full series, b; a zoom in the 54-83 period). Here we have to comment again that the reconstruction method is different than the one of the submitted paper (see comment below).

Differences for the 15 year window differences on the calculated trends outside the 54-83 period are less than 1%, with maximum differences at the late 60's ~3%. For the 30 year window maximum differences are in the order of ~2%., while for the 40 year window, maximum differences are less than 1%.

This particular exercise cannot be defined as an uncertainty assessment on the 1954-83 measurements, as reconstructed data cannot be used as a reference. Moreover, as SSR is much more sensitive than SD especially with respect to aerosol optical depth changes. So, in locations where the number of cloudless days is relatively high SD reconstruction tends to "smooth" the SSR variability, however the opposite can be said in cases with constant cloudiness where SD hours in certain days could be close to zero or zero while SSR is never zero.

Figures: Up to down; trends per decade using 15, 30 and 40 year windows, blue line: using the measured 1954-1983 data, orange line: using the 1954-83 reconstructed data, left panel: all years, right panel: zoom of left panel.

[Figure]

[Figure]

Based on all the discussion above we have decided to include a summary of most of the mentioned information on the new version of the manuscript. In addition, a paragraph mentioning:

'When trying to use such long term series it is evident that the data quality differs as instruments have been improved, quality assurance and quality control procedures have been standardized and finally the information flow on the day to day instrument performance issues are much more frequent in the recent years. More specific for the Athens station, after 1986 the instruments were calibrated or checked with a reference instrument in a yearly basis to identify changes in the calibration and drifts. As reported, the addition of diffuse irradiance measuring instruments provided the opportunity to improve also minute based measurement quality. Before 1986 the instruments reported in table 1 have been used. According with the log books there has been always a certain overlap when changing from one instrument to another. Reports mention that there were instrument drifts that have been corrected with no further information from 1953 to 1970. Instrument overlaps after 1986 were used to eliminate

possible instrument related offsets. However, instrument differences (e.g. thermal offset of PSP instrument compared with 8-48 pyranometer, Vignola et al., 2016) theoretically could have an effect in the order of 1-2 W/m$^2$ on the series continuation. In addition, the inclusion of diffuse radiation in the quality assurance tests after 1991 could have a major improvement on the newest data compared with the old ones. However, there is no hint that the improvement in quality control could have a systematic impact on SSR measured changes compared to the past, other than higher uncertainty on the integrated (monthly, yearly) values, by inclusion of "problematic" SSR minute or small period measurements that did not pass the quality controlled tests. For the 1953-1986 time series there is a number of publications that have been using the SSR-NOA time series. More specific: Macris, 1959, have used the 1954-1956 SSR measurements to identify the relationship of SSR and sunshine duration. Katsoulis and Papachristopoulos, 1978, have used the SSR data from 1960 to 1976 in order to calculate SSR statistics for daily, seasonal and yearly solar radiation levels for Athens, Greece. Notaridou and Lalas, 1979, have used the 1954-1976 SSR data in order to verify an empirical formula on global net radiation over Greece. Flocas, 1980 has used the 1961-1975 SSR time series to compare them with sunshine duration data for the same period. Kouremenos et al., 1985 have used the SSR data from 1955-1980 in order to try to correlate their changes with various atmospheric parameters. Zabara, 1986 has used the 1965-1980 time series to verify a developed method that calculated monthly solar radiation. Katsoulis and Leontaris, 1981, have used the 1960-1977 data to verify tools describing the solar radiation distribution over Greece. Finally, the percentages of errors reported in table 1, which are not directly linked with possible instrument drifts, can impact the SSR time series analysis. So results of measurements before 1986 have to be used with caution and accompanied by a report on the different level of uncertainties of the past and recent data.

In addition we have added a text summarizing the effect of the real and reconstructed 1954-1983 data to the overall calculated trends per decade based on the above figures. New figure 8 seen above in this comment was also replaced the current one mentioning the periods with possible instrument drifts.

Concerning the abstract and conclusions we have added sentences on the data quality:

Absolute year to year changes in SSR for the 1954-1983 measurement period have to be treated with caution as they can include uncertainties related with instrument exchange and undetected/uncorrected instrument drifts. However, long term (e.g. 25 year windows and more) calculated trends could only partly be affected by such uncertainties. If reconstructed series can be used as a hint for such drifts/steps, their level is not higher than 2% and only for very specific periods.

*P. 9, l. 31: how are SSR max and SDmax calculated?*

Maximum values of SSR and SD in use were theoretical extraterrastial and astronomical value accordingly. The uncertainties related with these parameters and their effect on the reconstruction method was tested by a new reconstruction method used. In this new method such assumptions were not included. See specific comment below.

*P. 12, figure 4: estimated SSR/SSRmax values (figures 2 and 3) show a typical 10- 20% spread around the fitting line. This is expected, since the used relationship takes into account*

*only cloud duration. All other effects (most of the aerosol direct effects, as well as most changes in cloud properties) can not be reproduced by the method. What is the uncertainty associated with these SSR estimates? Was this uncertainty considered in the trend analysis? Can these data be reliably used for trend analysis?*

In the initial submission we have used the Ångström related formula in order to calculate SSR and SD related functions. This method includes the theoretical SSR and SD maximum values in order to work that insert an uncertainty for such calculations. After the reviewer's comment we decided to replace this method with the one used by Sanchez-Lorenzo and Wild, 2012. One additional reason to test this method (as mentioned also in the paper) was the fact that monthly based calculated SD to SSR conversion functions had high uncertainty, linked with the very small SD/SDmax absolute variability especially for summer months.

In this new approach (Sanchez-Lorenzo and Wild, 2012) we did not use SSR and SD theoretical maxima in order to normalize the two factors but monthly anomalies of SSR and SD have been used for a common measuring period and then the monthly coefficients of the regression of SSR and SD anomalies were used in order to reconstruct the 1900-1953 time series.
The regression statistics of these monthly based SSR and SD anomalies analysis showed much better results from the Ångström method. As an example (and included in the new manuscript) statistics and graphs are shown below.

[Figure]

| month | Jan | Feb | Mar | Apr | May | Jun | Jul | Aug | Sep | Oct | Nov | Dec |
|-------|-----|-----|-----|-----|-----|-----|-----|-----|-----|-----|-----|-----|
| a | 22.47 | 34.99 | 39.53 | 46.65 | 57.88 | 44.67 | 51.87 | 46.23 | 34.28 | 28.02 | 27.32 | 23.10 |
| R | 0.842 | 0.895 | 0.887 | 0.840 | 0.799 | 0.757 | 0.773 | 0.572 | 0.812 | 0.888 | 0.916 | 0.842 |

According to Sanchez-Lorenzo and Wild, 2012, the correlation coefficients, here in the range of 0.75 to 0.91 can explain 65% to 82% of the variability of the SSR monthly anomalies. This additional verification analysis shows that the method used in this work is in line with Sanchez-Lorenzo and M. Wild (2012) that analysed 17 stations with very long term SDU series.

After having calculated the reconstructed series with this method we have compared the yearly and monthly SSR deviations with the ones calculated with the Angstrom method using

the yearly functions (initial submission). The results in yearly basis for all 1900-1953 period differ at a maximum 1%.

The agreement of these two results shows that in the case that SD measurements in the past have no particular quality issues, then SSR can be reconstructed with the 65-82% explained variability already mentioned.

Finally we have decided to keep the new method on the revised document and include the (yearly based) Ångström results as a verification. The inclusion of this method had a direct impact on all related figures 2, 3, 4, 5, 8 and tables describing trends that include the 1900-1953 period. As already reported the differences were small but still all the plots and tables have been replaced with the new ones calculated based on the Sanchez-Lorenzo and Wild (2012) method.

*Also, it is surprising that no significant signals of large volcanic eruptions (Agung in 1963, El Chichon in 1984, Pinatubo in 1991) are present in figure 4. A small SSR reduction in the early 90's, possibly related with Pinatubo, appears in figure 8; however, the minimum during 1990's in fig. 4 seems too late to be ascribed to Pinatubo (whose effect lasted for up to 2 years). Is there a possibe explanation?*

El Chichon date is linked with a 7% drop from 1983 to 1984. So we think that that it is at least partly visible when looking at the whole time series where year to year observations are more linked with cloud conditions.

Figure 4 in the previous manuscript included a wrong shift of one year that is why it did not exactly matched with figure 8. In addition, smaller differences raised from the fact that as normal period was the 1900-2012 while in the new document the 1984-2012. This was corrected, in addition to the (small) changes that were introduced with the change to the Sanchez–Lorenzo and Wild (2012) reconstruction method.

Here is a zoom of the 1987 – 1995 SSR for the four different seasons, Together with the stratospheric aerosol load calculated based on the ChArMEx AOD (Nabat et al., 2013) series.

[Figure]

[Figure]

We believe that the ~6% drop from 1990 to 1991-1993 shown for all seasons is a hint of the effect of the eruption on SSR data for the Athens station. However, as shown in figure 8 cloudiness for 1991 is also high, while is much lower for 1992 and 1993. Combined with the stratospheric AOD figure, seems that 1991 related decrease is also related with cloud increase while 92 and 93 one with the Pinatubo related aerosol effect.

*P.13, l.2: the graph also shows a clear decrease during 1950's.*

*This sentence has been restated*

*P.13, l.3: shows*

Typo has been corrected

*P. 13, table 2: why were these periods chosen? P. 13, l.*

The periods have bene chosen based on the data retrieval (reconstruction or measurements) (1900-1952 period and 1953 to 2012). And then measurement period has been divided in two 30 year periods in order to try to compare them (figure 6).

*19-22: does the trend determination and its statistical significance take into account uncertainties?*

This has been extensively discussed in a comment above.  A sentence has been added:
We have added:
It has to be noted that the trend determination and its statistical significance does not take into account measurement or SSR reconstruction related uncertainties, which are different for the different periods.
In addition to the new figure 8 and related comments and the comments on the reconstructed-actual 54-83 series related trends.

*P. 14, figure 5: this figure does not seem to support the choice of the periods used in table 2 for the trend calculations.*

The periods have been chosen based on different criteria (see comment *P. 13, table 2, above.* Figures like fig. 5 can be used for any reader to draw his conclusion on any SSR change under any time window.

*P. 17, fig. 8: the units for yearly mean SSR and total cloud cover are missing in the graph. The evolution of the yearly mean SSR does not seem to be coherent with the annual series of de-seasonalized SSR in figure 4 (the minimum in early 1990's does not seem to coincide with the minimum in mid 1990's in figure 4; the minimum in 1970 in figure 4 appears earlier in figure 8). Is there an explanation for that?*

The units are % deviations from normal for both SSR and clouds.
See comment above for differences in previous fig. 4 and 8.

*P. 18, l. 7: "presence of" may be removed*

It has been removed.

*P. 18, l. 8: figure 12 and the related discussion suggest that there is a long-term change in the number of cloudy days. Conversely, no significant change in the annual mean cloud cover appears. May this be taken as an indication of changes in cloud properties or distribution?*

Yes that is really interesting. We added a paragraph:

"Differences on the ratio of cloudless days shown in figure 12 and on the almost constant cloud octa variability shown in figure 8, is partly attributed to the different definition of a cloudless day that is based on the cloud radiative effect for fig. 12, and on observation of cloud percentage in the sky for fig. 8. However, this can also be an indication of changes in cloud properties (e.g. change in optically thin clouds that could have small radiation effect but are marked as cloudy conditions from the observer)."

*P. 18, l. 20: it may be emphasized that the clear sky selection criterion eliminates cases with high aerosol optical depth.*

We agree. A sentence has been added: "It may be emphasized that the clear sky selection criterion could possibly eliminate cases with very high aerosol optical depth".

*P. 21, l. 4-5: apparently, there is no stratospheric aerosol contribution in the ChArMEx AOD dataset. The large volcanic explosions are important events with an expected impact on SSR, and datasets which include these cases should be used. Please, explain more clearly what is the meaning of "..uses the trend and not the interannual variability which is not included in the global model that was used".*

The ChArMEx AOD (Nabat et al., 2013) accounts for tropospheric aerosols only, and does not include stratospheric AOD coming from large volcanic explosions. We agree that such explosions have an important impact on SSR. We have provided the evolution of stratospheric AOD in Athens (below), where the two main peaks are respectively due to El Chichon and Pinatubo eruptions. These two peaks are possibly associated with decreases in SSR in Athens (Figure 11 in the paper).

Concerning the second part of the comment, the trend in the ChArMEx AOD comes indeed from a global climate model which has no nudging towards reanalysis. Consequently it was impossible to deduce the interannual variability from this model, that is the reason why the ChArMEx AOD only accounts for the trend in AOD due to the decrease of anthropogenic emissions.

[Figure]

Stratospheric AOD over Athens

*P. 21, l.12-13: a change of almost a factor of 2 in the frequency of cloudless days seems to be non marginal. No evident effect appears on SSR in figure 8. However, trends in table 3 are calculated in periods separated around the years with minimum number of cloudless days. May part of the trend change in the two periods due to the long-term change of cloudless days/cloud properties (see also comment to p. 18, l.8)?*

Based on figure 12 there is a negative change in the number of cloudless days from 1970 to 1985 and a positive one from 1985 to 2000.Figure 7a (now deleted after the recommendation of reviewer 2) shows this effect in total (cloudless plus cloudy) SSR. It shows an ~ +5% change for the first period (e.g. blue line year 1977-78) and a ~0% change for the second period (e.g. blue line year 1992-93). However the problem is more complex as for cloudless days the AOD plays an important role and for cloudy days the cloud radiative effects also play a role always as a function of solar elevation which determines the SSR measurement absolute value.

*P. 22, l.25-P.23, l. 9: this discussion seems not fully consistent with the conclusions of the paper. For instance, Founda et al (2016) show that visibility is strongly related with AOD; and the paper highlights a possible role of aerosols in affecting SSR.*

The discussion on the visibility now transferred as last paragraph of the conclusions

(Purely mathematically speaking) "In this work and any other work that use SD to reconstruct SSR time series, reconstructed SSR is purely driven by actual sunshine duration changes. Founda et al., 2014 has presented the change of the SD since 1900. Using the measurements data we can calculate the % deviations for SD since 1900. These are in accordance with the reconstructed SSR. More or less the decrease of SSR from 1910 to ~ 1940 and the increase afterwards till ~1950 is also shown in SD. (Founda et al., 2014).

The visibility related study by Founda et al., 2016 shows the visibility variability since 1931. So a first difference is that the first 30 years are missing. A comparison of the SSR results in

Athens with visibility observations since 1931 (Founda et al., 2016) did not show any correlation among SSR and horizontal visibility. For the first part of the common dataset (1930-1959) the visibility decline is accompanied with a SSR increase. However from 1950 till today visibility shows a monotonical decrease. The steep visibility decrease from 1931 till the 90's is not accompanied by a relative SSR or SD decrease excluding individual sub-periods.

As already reported, simulated SSR is driven purely by changes in sunshine duration, in this case the SD variability in Founda et al., 2014 is almost constant after 1950 so SD also, can not be linked with the visibility reported decrease. Studying the literature for similar cases, similar conclusions have been drawn by Liepert and Kukla (1997) showing an SSR decrease over 30 years of measurements accompanied by a visibility increase and no significant changes in the cloud cover conditions, in Germany. This Athens SSR vs visibility relationship can be partly explained by the fact that: SSR and visibility have different response on cloud conditions, water vapor and rainfall and also by the fact that visibility is affected by aerosols only in the first few hundred meters above the surface, while SSR is affected by the columnar AOD, which in the case of Athens can be significantly different due to aerosol long-range transport in altitude (e.g. Saharan dust; Léon et al., 1999)."

**References**

Coulson, K.L.: Solar and Terrestrial Radiation: Methods and Measurements, Academic Press, 1975.

Drummond, A.J. and Roche, J.J.: Corrections to be applied to measurements made with Eppley (and other) spectral radiometers when used with Schott colored glass filters, Journal of Applied Meteorology, 4(6), pp.741-744, 1965.

Founda, D., Kalimeris A., Pierros F.: Multi annual variability and climatic signal analysis of sunshine duration at a large urban area of Mediterranean (Athens). Urban Climate, http://dx.doi.org/10.1016/ j.uclim.2014.09.008, 2014.

Founda, D., Kazadzis, S., Mihalopoulos, N., Gerasopoulos, E., Lianou, M. and Raptis, P.I.:Long-term visibility variation in Athens (1931–2013): a proxy for local and regional atmospheric aerosol loads. Atmospheric Chemistry and Physics, 16(17), pp.11219-11236, 2016.

Hulstrom, R. L.: "Solar resources", is the volume 2 in the Series: "Solar Heat Technologies: Fundamentals and Applications", edited by Charles A. Bankston, The MIT Press, Cambridge, 1989.

Léon, J.F., Chazette, P., and Dulac, F.: Retrieval and monitoring of aerosol optical thickness over an urban area by spaceborne and ground-based remote sensing, Appl. Opt., 38, 6918-6926, 1999.

Liepert B. and Kukla G., Decline in Global Solar Radiation with Increased Horizontal Visibility in Germany between 1964 and 1990, Journal of Climate, 2391-2401, September, 1997.

Nabat, P., Somot, S., Mallet, M., Chiapello, I., Morcrette, J.J., Solmon, F., Szopa, S., Dulac, F., Collins, W., Ghan, S. and Horowitz, L.W.,: A 4-D climatology (1979-2009) of the monthly tropospheric aerosol optical depth distribution over the Mediterranean region from a

comparative evaluation and blending of remote sensing and model products, Atmospheric Measurement Techniques, 6(5), p.1287, 2013.

Sanchez-Lorenzo, A., and Wild, M.: Decadal variations in estimated surface solar radiation over Switzerland since the late 19th century, Atmospheric Chemistry and Physics 12.18: 8635-8644, 2012.

Robinson, G. D. (1966), Some determinations of atmospheric absorption by measurement of solar radiation from aircraft and at the surface. Q.J.R. Meteorol. Soc., 92: 263–269. doi:10.1002/qj.49709239209

B. Katsoulis and E. Papachristopoulos, Analysis of solar radiation measurements at Athens Observatory and estimates of solar radiation in Greece, Solar Energy. Vol. 21, pp, 217-226, 1978

V. Notaridou and D. Lalas, The distribution of global and net radiation over Greece, Solar Energy Vol. 22, pp. 504-514

A. Flocas, Estimation and prediction of global solar radiation over Greece, Solar Energy, Vol. 24, pp 63-70, 1980

D. Kouremenos, K. Antonopoulos and E. Domazakis, Solar radiation correlations for Athens, Solar Energy Vol. 35, No. 3 pp 259-269, 1985

K. Zabara, Estimation of the global solar radiation in Greece, Solar & Wind Technology, Vol. 3, No. 4 pp 267-272, 1986

B. Katsoulis and S. Leontaris, The distribution over Greece of global solar radiation on a horizontal surface, Agricultural Methodology, 23, 217-229, 1981

G. Macris, Solar Energy and Sunshine hours in Athens, Greece, Monthly Weather review, January pp 29-32, 1959

García, R. D., Cuevas, E., García, O. E., Cachorro, V. E., Pallé, P., Bustos, J. J., Romero-Campos, P. M., and de Frutos, A. M.: Reconstruction of global solar radiation time series from 1933 to 2013 at the Izaña Atmospheric Observatory, Atmos. Meas. Tech., 7, 3139-3150, https://doi.org/10.5194/amt-7-3139-2014, 2014.

M. Anton, R. Roman, A. Sanchez-Lorenzo, J. Calbo, J.M. Vaquero, Variability analysis of the reconstructed daily global solar radiation under all-sky and cloud-free conditions in Madrid during the period 1887–1950, https://doi.org/10.1016/j.atmosres.2017.03.013, Atmospheric Research, 2017

Frank Vignola, Joseph Michalsky, Thomas Stoffel, Solar and Infrared Radiation Measurements, ISBN 9781439851906, CRC Press, 2012

---

## Editor Decision (ED1)

*and trends in*

**Long-term series of surface solar radiation at Athens, Greece**

S. Kazadzis[1,2], D. Founda[2], B. E. Psiloglou[2], H. Kambezidis[2], N. Mihalopoulos[2,3], A. Sanchez-Lorenzo[4], C. Meleti[5], P. I. Raptis[1,2], F. Pierros[2], P. Nabat[6]

[1] {Physikalisch-Meteorologisches Observatorium Davos, World Radiation Center (PMOD/WRC) Dorfstrasse 33, CH-7260 Davos Dorf, Switzerland}

[2] {Institute of Environmental Research and Sustainable Development, National Observatory of Athens, Greece}

[3] {Department of Chemistry, Univ. of Crete, Heraklion, Crete}

[4] {Instituto Pirenaico de Ecología, Consejo Superior de Investigaciones Científicas (IPE-CSIC), Zaragoza, Spain}

[5] {Physics Department, Aristotle University of Thessaloniki, Greece}

[6] {CNRM UMR 3589, Météo-France/CNRS, Toulouse, France}

Corresponding author: S. Kazadzis, kazadzis@noa.gr

**1 Abstract**

We present a long-term series of solar surface radiation (SSR) for the city of Athens, Greece. The SSR measurements were performed from 1954 to 2012, and before that (1900-1953) sunshine duration (SD) records have been used in order to reconstruct monthly SSR. Analysis from the whole dataset (1900-2012) mainly showed: Very small (0.02%) changes in SSR from 1900 to 1953, including a maximum decrease of 2.9% per decade in SSR from when taking in to account the 1910 to 1940 period, assuming a linear change in SSR. For the dimming period (1955-1980), a -2% change per decade has been observed, that matches various European long-term SSR measurement related studies. This percentage for Athens is in the lower limit, compared to other studies for the Mediterranean area. For the brightening period (1980-2012) we have calculated a +1.5% per decade, which is also in the lower limit of the reported positive changes in SSR around Europe. Comparing the 30-year periods (1954-1983 and 1983-2012) we have found a difference of 4.5%. However, measurements of the first 30 year period are associated with higher uncertainties than the second period, especially when looking at year to year changes. The difference was observed for all seasons except winter. Using an analysis of SSR calculations of all sky and clear sky (cloudless) conditions/days, we report that most of the observed changes in SSR after 1954 can be attributed partly to cloudiness and mostly to aerosol load changes.

**1 Introduction**

In the past decades surface solar radiation (SSR) and the transmission of the atmosphere have been of increasing interest because of the related impacts on climate. Most of the energy in the Earth-atmosphere system is introduced by solar radiation as it provides heating, which creates pressure gradients and ultimately wind, as well as it triggers water, carbon and oxygen cycles through evaporation and photosynthesis. These processes define the climatological conditions, and changes of incoming solar radiation rapidly affect the energy balance (Wild et al., 2015). Interest in the solar radiation changes has also been raised after the development of solar energy applications, which are continuously growing in number over the recent years. Changes in SSR have been recorded over the last century and can be caused either by natural events such as volcanic eruptions or human-related activities, mainly in polluted regions (Wild, 2016). At larger scales (thousands of years) changes in SSR might have been caused by changes in the Earth's orbit and Sun solar output (Lean, 1997; Ohmura, 2006).

Systematic continuous measurements of SSR were established in the middle of the 20th century at selected meteorological observatories. Solar variations have been investigated in several studies using ground based SSR measurements from various monitoring networks worldwide (e.g., Ohmura, 2009) and also by satellite-derived estimations (e.g. Kambezidis et al., 2010). Overall, most of these studies (Gilgen et al., 1998; Noris and Wild ,2009; Wild, 2009 and 2016 and references therein) have reported a worldwide decrease of solar incoming radiation in the period 1960-1985 (known as dimming period), followed by an increase (brightening period) thereafter. These changes were larger reported to be higher in more polluted and urban areas but have also been recorded in isolated regions such as the Arctic (Stanhill, 1995) and Antarctica (Stanhill and Cohen 1997). More Other recent studies have tried to distinguishinvestigated the effect of urbanization on global brightening and dimming, the local scale changes in polluted urban environments in the same time and found no marked differences among urban and rural SSR time series proved the existence of these trends independently of local sources (Tanaka et al., 2016) and Imamovic et al. (2016) investigated the correlation among these trends and urbanization proxies and found non-significant linkage. 
[revised manuscript text omitted]

---

## Author Response (AR2)

The authors would like to thank the reviewers and the editor Dr. Dulac for their efforts on improving the manuscript. We have taken into account the corrections of the editor and author reports.